# MEMORIZATION THROUGH THE LENS OF SAMPLE GRADIENTS

**Deepak Ravikumar**[1]*, **Efstathia Soufleri**[2], **Abolfazl Hashemi**[1], **Kaushik Roy**[1]
[1]Electrical and Computer Engineering, Purdue University, West Lafayette, IN 47907, USA
[2]Archimedes, Athena Research Center, Greece
{dravikum, abolfazl, kaushik}@purdue.edu, e.soufleri@athenarc.gr

## ABSTRACT

Deep neural networks are known to often memorize underrepresented, hard examples, with implications for generalization and privacy. Feldman & Zhang (2020) defined a rigorous notion of memorization. However it is prohibitively expensive to compute at scale because it requires training models both with and without the data point of interest in order to calculate the memorization score. We observe that samples that are less memorized tend to be learned earlier in training, whereas highly memorized samples are learned later. Motivated by this observation, we introduce Cumulative Sample Gradient (CSG), a computationally efficient proxy for memorization. CSG is the gradient of the *loss with respect to input samples*, accumulated over the course of training. The advantage of using input gradients is that per-sample gradients can be obtained with negligible overhead during training. The accumulation over training also reduces per-epoch variance and enables a formal link to memorization. Theoretically, we show that CSG is bounded by memorization and by learning time. Tracking these gradients during training reveals a characteristic rise–peak–decline trajectory whose timing is mirrored by the model's weight norm. This yields an early-stopping criterion that does not require a validation set: stop at the peak of the weight norm. This early stopping also enables our memorization proxy, CSG, to be up to *five orders of magnitude* more efficient than the memorization score from Feldman & Zhang (2020). It is also approximately *140×* and *10×* faster than the prior state-of-the-art memorization proxies, input curvature and cumulative sample loss, while still aligning closely with the memorization score, exhibiting high correlation. Further, we develop Sample Gradient Assisted Loss (SGAL), a proxy that further improves alignment with memorization and is highly efficient to compute. Finally, we show that CSG attains state-of-the-art performance on practical dataset diagnostics, such as mislabeled-sample detection and enables bias discovery, providing a theoretically grounded toolbox for studying memorization in deep [1].

## 1 INTRODUCTION

Deep learning has become the de facto standard across a wide array of machine learning tasks from generative models (Ho et al., 2020), and classification (Krizhevsky et al., 2009; Soufleri et al., 2024a), to reinforcement learning (Shakya et al., 2023). Despite their success, deep neural networks are prone to memorizing the training data. While some degree of memorization is needed for achieving generalization (Feldman, 2020), it has been shown that these models are also capable of memorizing completely random labels (Zhang et al., 2017). Understanding the mechanisms of memorization has thus emerged as a key research focus with broad implications for generalization (Bayat et al., 2025; Brown et al., 2021) and privacy (Ravikumar et al., 2024a).

Several methods have been proposed to measure memorization. Feldman (2020) offers a principled definition with strong theoretical and empirical support (Feldman & Zhang, 2020). However, it is computationally expensive. Their score quantifies memorization as the change in a model's output

---

*Corresponding Author
[1]Code link: https://github.com/DeepakTatachar/Sample-Gradient-Memorization

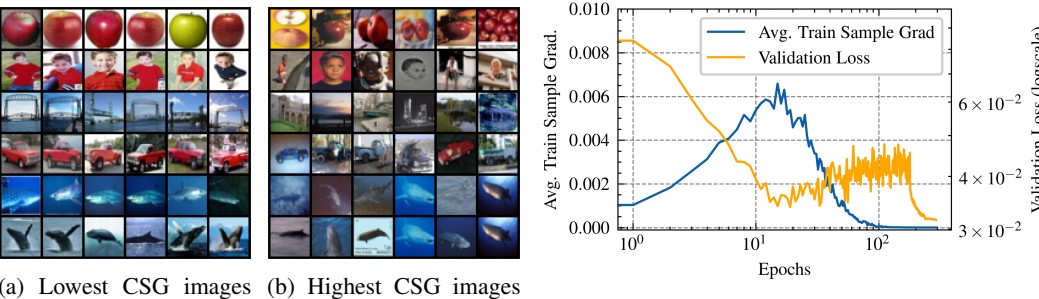

(a) Lowest CSG images from CIFAR-100.

(b) Highest CSG images from CIFAR-100.

(c) Average sample gradient and validation loss.

Figure 1: (a) Lowest CSG images from CIFAR-100 for 6 classes: captures examples prototypical for the class i.e. easy examples. (b) Highest CSG images from CIFAR-100 for the same 6 classes. (see Figure 16 for ImageNet): captures examples atypical for the class i.e. hard examples, likely memorized. (c) Plot of average train sample gradient for ResNet18 trained on CIFAR-100 and the corresponding validation set loss. The peak in sample gradient corresponds to the lowest validation loss (first descent of double descent). Thus, one can early stop without needing a validation set!

when a specific training sample is removed, requiring the training of $O(\text{dataset size})$ models. To address this, proxies such as model confidence (Carlini et al., 2019b), learning time (Jiang et al., 2021) and adversarial distance (Del Grosso et al., 2022) have been proposed. Yet, critical limitations persist among these alternatives. Methods relying on optimizer discrepancies (Agiollo et al., 2024) or influence functions (Pruthi et al., 2020) often incur high computational costs due to k-fold validation and parameter-space single-sample gradients, respectively. Similarly, metrics based on gradient variance, such as VoG (Agarwal et al., 2022), can misidentify consistently hard samples and are ad hoc metrics lacking theory linking them to memorization. Leveraging training dynamics has led to the development of further proxies, including loss sensitivity (Arpit et al., 2017), forgetting frequency (Toneva et al., 2019), the C-score (Jiang et al., 2021), and cumulative loss (Ravikumar et al., 2025a). Other approaches examine loss landscape sharpness (Krueger et al., 2017) and input loss curvature (Garg et al., 2024), with Ravikumar et al. (2024a) establishing theoretical links to memorization and privacy. While these proxies offer valuable insights, many fail to capture critical properties of memorization, such as its bi-modality (Lukasik et al., 2023). Developing a stronger theoretical foundation is therefore essential. We observe that less-memorized samples tend to be learned earlier in training, whereas highly memorized samples are learned later (see Fig. 2). Motivated by this, we propose a new proxy, Cumulative Sample Gradient (CSG). *CSG is defined as the gradient of the loss with respect to the input data, aggregated across training.* We establish a theoretical framework linking CSG, memorization, and learning time, and validate it experimentally by showing that CSG is highly correlated with memorization (Feldman & Zhang, 2020).

Interestingly, the average per-sample gradient follows a rise–peak–decline trajectory (see Figure 1c), a pattern mirrored by the network's weight norm. We show this arises because the per-iteration weight norm bounds the sample gradient. Crucially, the peak coincides with the minimum validation loss (the first minimum of double descent, Section 4.3). This yields a simple early-stopping criterion: stop at the peak of the weight norm. Using this property, we introduce Sample Gradient Assisted Loss (SGAL) a proxy that further improves alignment with memorization. By early stopping, CSG and SGAL achieve up to *five orders of magnitude* speedup over estimation of memorization by (Feldman & Zhang, 2020), and are substantially faster than prior state-of-the-art proxies, including curvature (Garg et al., 2024) (up to *140×*) and CSL (Ravikumar et al., 2025a) (max *10×*, avg. *5×*). Finally, we demonstrate that CSG effectively discovers dataset bias and achieves state-of-the-art performance in identifying mislabeled samples. In summary, our contributions are:

- **Theoretical foundations:** We develop a theoretical framework linking CSG to memorization and learning time, offering up to 5-orders-of-magnitude speedup over memorization score Feldman & Zhang (2020), $140\times$ and $10\times$ speedup than previous state-of-the-art memorization proxies such as Curvature (Garg et al., 2024) and CSL (Ravikumar et al., 2025a), respectively.

- **Practical impact and insights:** CSG achieves state-of-the-art performance for identifying mislabeled samples. Our experiments validate that sample gradients enable early stopping without a validation set, and reveal biases in training data.

## 2 NOTATION

We denote distributions using bold capital letters $\mathbf{V}$, random variables sampled from them as italic small letters $v$ for scalars, $\vec{v}$ for vectors, and capital letters $V$ for matrices. Consider a learning problem, where the task is learning the mapping $f : \boldsymbol{x} \mapsto y$ where $\vec{x} \sim \mathbf{X} \in \mathbb{R}^n$ and $y \sim \mathbf{Y} \mid \mathbf{X} \in \mathbb{R}$. Let this task be learnt using a dataset $S = (\vec{z}_1, \vec{z}_2, \ldots, \vec{z}_m) \sim \mathbf{Z}^m$ consisting $m$ samples, where each sample $\vec{z}_i = (\vec{x}_i, y_i) \sim \mathbf{Z}$. Our analysis also makes use of a leave one out dataset denoted by $S^{\backslash i} = (\vec{z}_1, \ldots, \vec{z}_{i-1}, \vec{z}_{i+1}, \ldots, \vec{z}_m)$, which is $S$ with the $i^{th}$ sample removed. In this paper, we consider a deep neural network with $q$ layers. Its parameters at iteration $t$ are denoted by $\vec{w}_t = \left[ \vec{w}_t^{(1)}, \vec{w}_t^{(2)}, \ldots, \vec{w}_t^{(q)} \right] \sim \mathbf{W}$. Here, $t$ indexes the optimization iteration, and $\vec{w}_t^{(k)}$ collects the parameters of layer $k$. We use a flattened representation of the parameters. For the $k$-th layer (with input dimension $d_{k-1}$ and output dimension $d_k$), the weights are represented as the row vector $\vec{w}_t^{(k)} = \left[ w_{t,1,1}^{(k)} \quad w_{t,1,2}^{(k)} \quad \cdots \quad w_{t,d_k,d_{k-1}}^{(k)} \right]$, which lists the entries $w_{t,i,j}^{(k)}$ of the (unflattened) weight matrix for $i = \{1, \ldots, d_k\}$ and $j = \{1, \ldots, d_{k-1}\}$. Let $g_S^p \sim \mathbf{G}_S$ to denote the function learned by the neural network. The neural net is trained by the application of a possibly random training algorithm $\mathcal{A}$ such as SGD and $p \sim \mathbf{P}$ denotes the randomness of the algorithm. For example in mini-batch SGD there are two sources of randomness, (a) the choice of mini-batch (b) model initialization. Here, $p$ captures (a). To evaluate model performance, we use the loss at a sample $\vec{z}_i$, defined as $\ell(g, \vec{z}_i) = \ell(g(\vec{x}_i), y_i)$. Since the function learned at iteration $t$, $g_t$, is fully parameterized by $\vec{w}_t$, we use $\ell(\vec{w}_t, \vec{z}_i) = \ell(g_t, \vec{z}_i)$ interchangeably. Typically, we are interested in the loss of $g$ over the entire data distribution, called the population risk, which is defined as $R(g) = \mathbb{E}_z[\ell(g, \vec{z})]$. In practice we evaluate the empirical risk as follows $R_{\text{emp}}(g, S) = \frac{1}{m} \sum_{i=1}^m \ell(g, \vec{z}_i), \vec{z}_i \in S$.

## 3 RELATED WORK AND BACKGROUND

Understanding memorization has broad implications, ranging from generalization (Brown et al., 2021; Zhang et al., 2017; 2021), unlearning (Kodge et al., 2025; 2024; Kurmanji et al., 2023), and privacy (Dwork et al., 2006; Soufleri et al., 2024b), to identifying mislabeled and rare subpopulation examples (Pleiss et al., 2020; Maini et al., 2022; Ravikumar et al., 2023). While many notions of memorization exist from exact (Kandpal et al., 2022) and $k$-eidetic (Carlini et al., 2021) to extractable memorization (Nasr et al., 2023) in this paper we focus on stability-based memorization, also referred to as counterfactual memorization, introduced by Feldman (2020). We hereafter refer to this as the memorization score. Formally, the memorization score of the $i^{\text{th}}$ sample $\vec{z}_i = (\vec{x}_i, y_i)$ for a network $g_S^p$ trained on dataset $S$ using algorithm $\mathcal{A}$ (with randomness $p$) is defined as:

$$\text{mem}(S, \vec{z}_i) = \Pr[g_S^p(\vec{x}_i) = y_i] - \Pr[g_{S^{\backslash i}}^p(\vec{x}_i) = y_i] \tag{1}$$

The term "stability-based" comes from its connection to uniform stability, a concept central to this work. Formally, an algorithm $\mathcal{A}$ is $\beta$-uniform stable (Kearns & Ron, 1997) if for some $\beta > 0$ :

$$\left| \ell(g_S^p, \vec{z}) - \ell(g_{S^{\backslash i}}^p, \vec{z}) \right| \leq \beta, \quad \forall i \in 1, \cdots, m. \tag{2}$$

An example of such an algorithm is Stochastic Gradient Descent (SGD) (Hardt et al., 2016). Proofs of its stability and convergence typically rely on a set of conditions we refer to as the SGD Convergence Assumptions: $\rho$-Lipschitz continuity (Eq. 8), $\Gamma_v$-bounded variance (Eq. 9), unbiased gradients, and a decreasing learning rate $\eta_t$ at each iteration $t$. Under these assumptions, SGD is guaranteed to converge in gradient norm (Ghadimi & Lan, 2013) (see Appendix A). Some results additionally require an $L$-bounded loss satisfying $0 \leq \ell \leq L$. While stability-based memorization provides a principled definition, it is prohibitively expensive to compute, as previously discussed. To overcome this, researchers have proposed proxies such as model confidence (Carlini et al., 2019b), learning time (Jiang et al., 2021), and adversarial distance (Del Grosso et al., 2022). Other methods leverage training dynamics such as loss sensitivity (Arpit et al., 2017), forgetting frequency (Toneva et al., 2019), the C-score (Jiang et al., 2021), and cumulative loss (Ravikumar et al., 2025a). More recently approaches study the loss landscape, e.g., sharpness (Krueger et al., 2017) and input loss

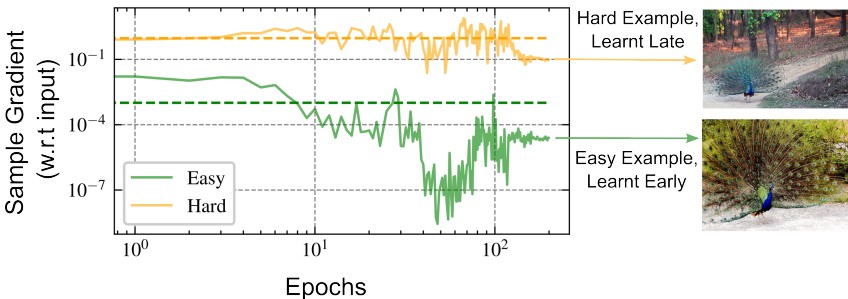

Figure 2: Visualizing Input Gradient Norms for Peacock-Class Samples in ImageNet. We plot the input gradient norm (for ResNet18) for one easy and one hard peacock-class training example. Solid lines show individual sample norms; dashed shows the average for the sample over training. The easy example is learned early, while the hard example is learned later and memorized more.

curvature (Garg et al., 2024; Ravikumar et al., 2024a). Alternative proxies relying on optimizer discrepancies (Agiollo et al., 2024) or influence functions (Pruthi et al., 2020) have also been proposed; however, these often incur high computational costs due to requirements for k-fold validation or parameter-space gradient calculations. Similarly, metrics based on gradient variance, such as VoG (Agarwal et al., 2022), focus on fluctuation rather than accumulation, which can misidentify consistently hard samples. Finally, while memorization in language models has been analyzed via exact sequence matching (Leybzon & Kervadec, 2024), such methods often lack a theoretical framework linking input gradient dynamics to formal memorization definitions. In contrast, our work investigates sample gradients (not weight but input) across training as a theoretically grounded proxy for memorization.

## 4 CUMULATIVE SAMPLE GRADIENT (CSG)

To develop intuition and motivate the introduction of CSG, we analyze how the input loss gradients evolve during training for two samples. Consider an "easy" and a "hard" example from the same class (peacock) in the ImageNet dataset (Russakovsky et al., 2015). Figure 2 plots the norm of the gradient of the *loss w.r.t. input* (sample loss gradient) vs epochs. This captures how well the model fits each sample over training. The easy sample, learned early, shows a rapid drop in gradient that stays low. The hard sample, by contrast, maintains a high gradient much longer. This contrast highlights how tracking the average sample gradient during training helps distinguishing between easy and hard examples. Traditional metrics like learning and forgetting time rely on thresholds and can be noisy e.g., a sample may appear learned, then forgotten, then relearned. To address this, we introduce cumulative sample gradient (CSG), which smooths such fluctuations and provides a more reliable signal. Hard examples tend to be memorized; easy ones are likely generalized which is captured by CSG (see Figures 1a, 1b, 16a and 16b). CSG for a sample $\vec{z}_i = (\vec{x}_i, y_i)$ is defined as:

$$\text{CSG}(\vec{z}_i) = T_{max} \cdot \mathbb{E}_R \left[ \|\nabla_{x_i} \ell (\vec{w}_R)\|_2^2 \right] \approx \sum_{t=0}^{T_{max}} \|\nabla_{x_i} \ell(\vec{w}_t)\|_2^2 \quad (3)$$

where $T_{max}$ is the total number of iterations of SGD[2]. For convenience, we introduce the concept of the sample learning condition as $\mathbb{E}_R \left[ \|\nabla_{x_i} \ell (\vec{w}_R)\|_2^2 \right] \leq \tau$, thus learning time $T_{z_i}$ can be defined using the sample learning condition as

$$T_{z_i} = \min_T \left\{ T : \mathbb{E}_R \left[ \|\nabla_{x_i} \ell (\vec{w}_R)\|_2^2 \right] \leq \tau \right\} \quad (4)$$

where the *gradient is with respect to the input* $\vec{x}_i$. Here, $R < T$ is a discrete random variable sampled non-uniformly from the set $\{1, 2, \ldots, T_{max}\}$. This borrows from the SGD convergence result of Ghadimi & Lan (2013). We interpret the definition as follows: a sample is deemed learned when the expected sample gradient drops below a predefined threshold $\tau$.

---

[2]Note that the approximation is justified by the law of large numbers.

## 4.1 CSG and Learning Time

Before stating our main result, we recall two preliminaries. First, consider SGD with update rule $\vec{w}_{t+1} = \vec{w}_t - \eta_t \widetilde{\nabla}_{w_t}\ell(\vec{w}_t, \vec{z}_i)$, where $\eta_t$ is the learning rate and $\widetilde{\nabla}_{w_t}\ell$ is an unbiased stochastic gradient estimator. Following Ghadimi & Lan (2013), a decreasing step-size schedule is required to ensure convergence, such that the sum $\eta = \sum_{t=0}^{T-1}\left(\eta_t - \frac{\rho}{2}\eta_t^2\right) < \infty$, with $\rho$ denoting the Lipschitz constant (Eq. 8). Second, building on Ravikumar et al. (2025a), we show that the input gradient is bounded by the weight norm and the weight-gradient norm. Since the weight-gradient norm is known to converge during training (Ghadimi & Lan, 2013), and the input norm remains fixed, the weight norm effectively bounds the input gradient. This is a re-framed version from Ravikumar et al. (2025a), but helps build towards our result. Formally we state the lemma below

**Lemma 4.1.** *For any neural network without a skip connection at the first layer, the Frobenius norm of the gradient of the loss $\ell$ with respect to a nonzero input $\vec{x}_i$ for sample $\vec{z}_i = (\vec{x}_i, y_i)$ is bounded by the norm of the network weights $\vec{w}_t$:*

$$\|\nabla_{x_i}\ell(\vec{w}_t, \vec{z}_i)\|_F \leq \|\vec{w}_t\|_F \|\nabla_{w_t}\ell(\vec{w}_t, \vec{z}_i)\|_F / \|\vec{x}_i\|_F , \qquad (5)$$

The lemma also holds for a mini-batch $X_b$. In which case, the constant becomes $1/\|\vec{x}_i\|_F$ becomes $\|(X_b^\top)^+\|_F/s_P$ where $s_P$ is the smallest singular value of $P = X_b^\top(X_b^\top)^+$ and $^+$ denotes pseudo-inverse. Using this general version (see proof in Appendix C.1), we define at each iteration $t$ the factor $\kappa_g^t = \|\vec{w}_t\|_F\|(X_b^\top)^+\|_F/s_P$ and $\kappa_m = \max_t\left(\kappa_g^t\right)^2$. We are now ready to state our main result.

**Theorem 4.2** (CSG is bound by Learning Time). *Consider a deep network without skip connections in the first layer, trained by SGD. Suppose the SGD convergence assumptions hold, the loss is $L$-bounded and SGD is $\beta$-stable. Then there exist constants $C_1, C_2 > 0$ such that the expected cumulative sample gradient (CSG) and expected learning time $T_{z_i}$ of a sample $\vec{z}_i$ satisfy:*

$$\mathbb{E}_{p,z_i}[\mathrm{CSG}(\vec{z}_i)] \leq C_1 \, \mathbb{E}_{p,z_i}[T_{z_i}] + C_2, \qquad (6)$$

**Sketch of Proof.** The proof establishes a bound on input gradient norms by adapting convergence results from randomized SGD but in the input space. It then connects this to the learning condition. By leveraging leave-one-out analysis and the $\beta$-stability of SGD, we separate the memorization term. Combining these pieces yields the final result. The proof is provided in Appendix C.2.

**Understanding the terms.** In the result, $\eta_s$ denotes the maximum learning rate during training, $T_{\max}$ the maximum number of iterations, $\rho$ the Lipschitz constant, and $\Gamma_v$ the variance bound. With these definitions recalled, the constants are

$$C_1 = \left[\tau - \frac{\kappa_m^2\eta_s\rho\Gamma_v^2}{2\eta}\right], \quad C_2 = \frac{\kappa_m\beta}{\eta} + \frac{\kappa_m^2\,\eta_s\,T_{\max}\,\rho\,\Gamma_v^2}{2\eta} \geq 0,$$

the constant $C_1$ depends on the problem-defined threshold $\tau$, which can be chosen so that $C_1$ is positive. The absolute value of $\tau$ is not essential; rather, the role of $\tau$ is to help provide theoretical connection. $C_2$ is determined by parameters related to $\beta$-stability and assumptions from SGD, and is therefore a positive fixed quantity once the network, and data are specified.

**Interpreting the Theory.** Theorem 4.2 can be interpreted as follows: *for any group or subset of training samples, their CSG scores are bounded by the group's learning time.* To see this, consider the subset $U(T) = \{\vec{z}_i : T_{z_i} \leq T\}$, which contains all samples learned within a time threshold $T$. For any randomly chosen subset $U$ drawn from the same distribution as the full dataset, we can define the corresponding threshold as $T = \max\{T_{z_i} : \vec{z}_i \in U\}$. Since Equation 6 holds for all such subsets, it follows that the average CSG within a subset is bounded in proportion to its average learning time. Finally, this can be validated in theory by grouping samples and observing a linear relationship between CSG and learning time, which is performed in Section 5.1. The theory also predicts that large cumulative input gradients must result in either very large learning times or samples that are never learned such as mislabeled examples. This is indeed the case, by thresholding on CSG we achieve state-of-the-art performance in mislabeled sample detection (see Sec. 5.4).

## 4.2 CSG and Memorization

**Theorem 4.3** (CSG is bound by Memorization). *Consider a deep network without skip connections in the first layer, trained by SGD. Suppose the SGD convergence assumptions hold, the loss is $L$-*

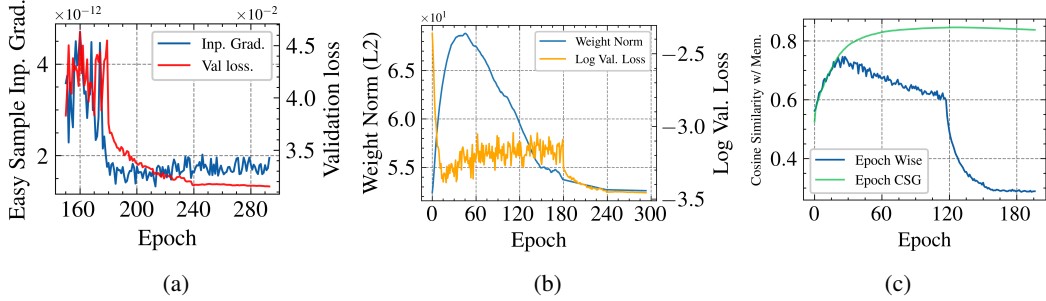

Figure 3: (a) Input-gradient norm over training epochs for the 100 easiest examples (lowest CSG) in CIFAR-100. The curve closely follows the second descent in validation loss (shown in red), highlighting their correlation. (b) Validation loss and weight norm plotted together. The first descent in the loss aligns with the peak in weight norm, reinforcing its connection to the double-descent. (c) Similarity between input gradients and memorization scores across epochs (blue), along with the cumulative similarity (green). The CSG similarity (green) plateaus around the first descent point.

*bounded and SGD is $\beta$-stable. Then, memorization bounds cumulative sample gradient as*

$$\mathbb{E}_{z_i}[\text{CSG}(\vec{z_i})] = O(\mathbb{E}_{z_i}[\text{mem}(\vec{z_i})]) \tag{7}$$

**Sketch of Proof.** The proof derives both a lower and an upper bound on the learning time using convergence properties of SGD and the learning condition. We eliminate learning time using the two bounds, to get an quadratic in terms of the CSG, where the coefficients depend on sample memorization. Analyzing this quadratic reveals that it's roots provide bounds on the CSG, showing CSG is linearly bound to memorization. Full proof is available in Appendix C.3.

**Interpreting the Theory.** The coefficient for $O$-notation, is proportional to $b/2a$ where $a$ and $b$ are the second and first order coefficients of the quadratic (See Eq. 52). Similar to Theorem 4.2, Theorem 4.3 applies to every set $U(m) = \{\vec{z_i} : \text{mem}(\vec{z_i}) \leq m\}$, where all samples have memorization bounded by $m$. For such subsets, the cumulative sample gradient (CSG) is linearly bounded in memorization. In other words, *groups or subsets with lower memorization tend to have lower CSG, while subsets with higher memorization are likely to exhibit higher CSG.*

**Remark on Assumptions.** Uniform stability of SGD has been demonstrated in prior work (Hardt et al., 2016), lending support to our stability assumption (Eq. 2). The Lipschitz continuity assumption is justified by results from Virmaux & Scaman (2018), where they derive general upper bounds on the Lipschitz constant for differentiable deep models. Assumptions on an unbiased gradient estimator and bounded gradient variance are standard in optimization theory (Aketi et al., 2024; Ghadimi & Lan, 2013). While we assume $L$-bounded loss, the proof can be potentially extended to cross entropy loss which is unbounded using result A.4 from Ravikumar et al. (2024a). Additionally, most architectures such as ViTs, ResNets and VGGs don't have a skip connection from the input. *Key Takeaways. If the theory bounds are tight, we expect the following: (1) learning time to exhibit a linear relationship with CSG, (2) memorization to follow a linear relation with CSG.*

### 4.3 SGAL AND LINKS TO DOUBLE DESCENT

**Peaky trajectory.** Figure 1c shows that the average sample gradient norm rises from a low baseline, peaks sharply, and then declines. The weight norm follows the same trend (Fig. 3b), since it upper bounds the sample gradient (Lemma 4.1). The weight norm is shaped by two opposing forces: performance loss pushes it upward to fit the data, while $\ell_2$ weight decay and SGD's implicit bias toward minimum-norm solutions (Park et al., 2023) pull it downward. Interestingly we also see this when using the Adam, AdamW, Adagrad, and RMSProp optimizers (see Appendix B.4 and Figures 10 and 11). Furthermore, we demonstrate consistency across different architectures specifically ViT (ViT-M-16/256) and ResNet50 in Figures 12a and 12b respectively. These results provide evidence that this phenomenon is robust to variations in both architecture and optimizer.

**Connection to double descent.** Double descent refers to the phenomenon where test loss decreases, then increases, and decreases again as a function of iterations, parameters, or dataset size. Here, we focus on iterations. This behavior is evident in Fig. 1c, where the test loss first drops, then plateaus,

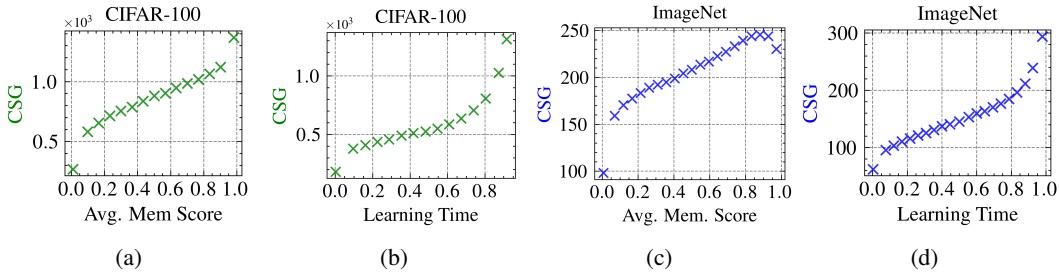

Figure 4: (a) Plots memorization score vs CSG on CIFAR-100 for Inception model (same arch. as Feldman & Zhang (2020)) (b) Plots learning time vs CSG for CIFAR-100 (Inception) (c) Plots memorization score vs CSG on ImageNet for ResNet50 (same arch. as Feldman & Zhang (2020)) (d) Plots learning time vs CSG for ImageNet.

rises, and finally drops again. Classical double-descent studies identify the first validation-loss minimum as the boundary between interpolation and generalization regimes (Poggio et al., 2019; Schaeffer et al., 2024). We observe that average CSG gradient peak lands on that same boundary (see Figure 1c). Additionally, at the peak, sample gradients are maximally aligned with memorization scores Feldman & Zhang (2020) (see Figure 3c), providing more evidence for the link.

**Practical takeaway.** Let's consider what happens under strong weight penalty. Strong penalty pushes the weight norm down immediately during training. Upon further training the performance loss pushes it back up, and finally SGD-bias drags it down. While weight decay is plays an important role, we found that for standard training recipes, one can stop at peak weight norm. Ideally, monitoring both sample gradient and weight norm is recommended (see Figures 10 and 11 for detailed effect of weight decay).

**SGAL.** In Figure 3c, we show the similarity between the input gradients and the memorization score across epochs (in blue), as well as the cumulative similarity up to each epoch (in green). We observe that CSG plateaus around the first descent point, or shortly after. This suggests that we can avoid training on the full dataset and still achieve strong similarity with memorization, significantly lowering compute costs. Motivated by this, we introduce sample Gradient-Assisted early stopping with accumulated Loss (SGAL), a practical proxy for memorization. SGAL accumulates sample loss until the gradient indicates an optimal stopping point, on average allowing training to stop after 10–30% of epochs and yielding a 3–10× efficiency gain. The theory behind SGAL's link to memorization is justified using the results from Ravikumar et al. (2025a). Additionally, input gradients also identify the second descent without a validation set, by analyzing gradients of the easiest examples (low CSG). Figure 3a shows this on CIFAR-100, where gradient norms of the 100 easiest examples align closely with validation loss.

## 5 EXPERIMENTS

### 5.1 VALIDATING THEORY

**Experiment.** Here, we examine how well our theory (i.e. Theorem 4.2 and 4.3) holds in practice. We conduct the experiment by training Inception (Szegedy et al., 2016) model on CIFAR-100 (Krizhevsky et al., 2009) and ResNet50 (He et al., 2016) on ImageNet (Russakovsky et al., 2015), same as Feldman & Zhang Feldman & Zhang (2020). For memorization scores, we utilize the precomputed memorization scores from Feldman & Zhang (2020). Since the theorems apply to groups of samples (i.e. under expectation), we grouped data by the x-axis metric and calculated average scores for each bin. We plot a binned scatter plot of these metrics (see Appendix B.1 for more details).

**Results.** The results are visualized in Figures 4a, 4b, 4c and 4d. Figures 4a and 4c plots the the memorization score from Feldman & Zhang (2020) against CSG for CIFAR-100 and ImageNet respectively. Figures 4b and 4d plots learning time vs CSG for CIFAR-100 and ImageNet respectively.

**Takeaways.** Theorem 4.2 predicts a linear bound between learning time and the sample gradient, and Theorem 4.3 also predicts a linear relationship with memorization. These predictions are sup-

Table 1: Cosine similarity (CS) and Pearson correlation (Corr.) between CSG and baselines with memorization (Feldman & Zhang, 2020), and normalized compute cost. † Computationally infeasible to scale to ImageNet.

| Method | Compute | CIFAR-100 | | ImageNet | |
|---|---|---|---|---|---|
| | | CS | Corr. | CS | Corr. |
| Final Sample Loss | $1\times$ | 0.24 | 0.17 | 0.63 | 0.49 |
| Forget Freq. (Toneva et al., 2019) | $1\times$ | 0.76 | 0.59 | 0.49 | 0.04 |
| Loss Sensitivity Arpit et al. (2017) | $1\times$ | 0.81 | 0.76 | 0.49 | 0.17 |
| CSL (Ravikumar et al., 2025a) | $1\times$ | *0.87* | *0.79* | *0.79* | *0.64* |
| VoG All Ckpts (Agarwal et al., 2022) | $1\times$ | 0.80 | 0.67 | 0.60 | 0.23 |
| VoG Last 5 Ckpts (Agarwal et al., 2022) | $1\times$ | 0.33 | 0.23 | 0.58 | 0.32 |
| GraNd (Paul et al., 2021) | $7.8\times$ | 0.81 | 0.36 | † | † |
| Curv. (Garg et al., 2024) | $14\times$ | 0.69 | 0.49 | 0.62 | 0.33 |
| SAMIS (Agiollo et al., 2024) | $24\times$ | 0.53 | 0.69 | † | † |
| TracIn (Pruthi et al., 2020) | $26\times$ | 0.83 | 0.71 | † | † |
| **SGAL (Ours)** | $\mathbf{0.1 - 0.3\times}$ | 0.86 | 0.77 | 0.78 | 0.62 |
| **CSG (Ours)** | $\mathbf{0.1 - 0.3\times}$ | 0.84 | 0.72 | 0.71 | 0.52 |

Table 2: Calibration results (mean $\pm$ std) for CIFAR-100 ResNet18 checkpoints

| Metric | Ours | Val. Loss Stop | Last Epoch |
|---|---|---|---|
| Accuracy | $0.6306 \pm 0.0103$ | $0.6262 \pm 0.0096$ | $0.7492 \pm 0.0046$ |
| ECE $\downarrow$ | $0.1382 \pm 0.0262$ | $0.1260 \pm 0.0083$ | $0.1017 \pm 0.0031$ |
| MCE $\downarrow$ | $0.2721 \pm 0.0458$ | $0.2352 \pm 0.0203$ | $0.2794 \pm 0.0149$ |
| MSCE $\downarrow$ | $0.4208 \pm 0.1620$ | $0.3257 \pm 0.0355$ | $0.4427 \pm 0.0410$ |
| UCE $\downarrow$ | $2.0821 \pm 0.4648$ | $1.8667 \pm 0.1424$ | $2.1839 \pm 0.0840$ |

ported by experimental results shown in Figures 4a, 4b, 4c, and 4d, which empirically confirm these trends. There is a slight deviation from the predicted linear relations at high learning-time and memorization respectively. This is likely due to the theoretical assumption of a uniformly bounded loss. In practice, the cross-entropy loss is not uniformly bounded. This issue is also discussed in detail by Ravikumar et al. (2024a), where accounting for subpopulation based loss bound improved the match between theory and practice.

## 5.2 SIMILARITY WITH MEMORIZATION

**Experiment.** We train Inception on CIFAR-100 and ResNet50 on ImageNet, computing memorization proxies and comparing them to memorization scores from Feldman & Zhang (2020) using cosine similarity and Pearson correlation. We also evaluate L2 adversarial distance and MIA performance (Appendix A.1), relating them to both memorization scores and CSG (see Appendix Figures 7–9b). Additional setup details are in Appendix B.2.

**Results.** Table 1 shows the similarity with memorization score of each of the proxies: CSG, SGAL, CSL (Ravikumar et al., 2025a), curvature (Garg et al., 2024), final sample loss, loss sensitivity (Arpit et al., 2017), forgetting frequency (Toneva et al., 2019), SAMIS (Agiollo et al., 2024), VoG (Agarwal et al., 2022), TracIn (Pruthi et al., 2020), and GraNd (Paul et al., 2021). It also lists the normalized compute cost for each method. Figure 7 shows that both CSG and Feldman & Zhang (2020)'s memorization scores follow similar trends with adversarial distance. Additionally, Figures 9a and 9b show the same using MIA performance (see Appendix B.7 for details on MIA). Additionally, we provide the same set of results when using Adam (Kingma, 2014), AdamW (Loshchilov & Hutter, 2019), RMProp (Hinton et al., 2012) and Adagrad (Duchi et al., 2011) optimizers in the Appendix Table 6.

**Takeaways.** CSG and SGAL demonstrate a strong correlation with memorization and serve as highly efficient proxies. SGAL achieves between 97–99% (Table 1) of the correlation obtained by the best proxy (CSL) while requiring only 10%-30% of the computational cost. CSG and SGAL are approximately $140\times$ faster than curvature, up to $10\times$ faster than CSL, and five orders of magnitude faster than memorization score (see cost analysis in Appendix B.5).

Table 3: Mean ± standard deviation of MIA performance over two runs for different stopping points. Lower values indicate better privacy.

| MIA Method | AUROC ↓ | | | Balanced Accuracy ↓ | | |
| --- | --- | --- | --- | --- | --- | --- |
| | Ours | Val. Loss Stop | Last Epoch | Ours | Val. Loss Stop | Last Epoch |
| Curvature Ravikumar et al. (2024b) | 59.42 ± 5.83 | 60.67 ± 4.06 | 85.52 ± 0.45 | 56.55 ± 4.16 | 57.41 ± 2.84 | 76.63 ± 0.26 |
| LiRA Carlini et al. (2022) | 55.98 ± 5.34 | 56.98 ± 4.93 | 85.48 ± 0.18 | 54.36 ± 3.90 | 55.09 ± 3.70 | 78.05 ± 0.24 |
| Cal. Loss Watson et al. (2022) | 67.02 ± 5.52 | 68.63 ± 5.23 | 70.94 ± 0.22 | 61.14 ± 3.12 | 62.34 ± 3.25 | 62.36 ± 0.16 |
| MAST Sablayrolles et al. (2019) | 70.70 ± 6.84 | 72.03 ± 4.89 | 78.56 ± 0.25 | 64.10 ± 4.94 | 65.06 ± 3.46 | 68.19 ± 0.26 |
| Yeom et al. Yeom et al. (2018) | 57.57 ± 4.72 | 58.27 ± 3.31 | 80.75 ± 0.16 | 55.91 ± 3.83 | 56.47 ± 2.70 | 74.23 ± 0.23 |

Table 4: AUROC (mean ± std over 3 seeds) for mislabeled-sample detection on CIFAR-10 and CIFAR-100 at five noise levels. Best values per column are in **bold**.

| Dataset | Method | 5% Noise | 10% Noise | 20% Noise | 25% Noise | 30% Noise |
| --- | --- | --- | --- | --- | --- | --- |
| CIFAR-10 | CL (Northcutt et al., 2021) | 0.8874 ± 0.0019 | 0.8551 ± 0.0030 | 0.7169 ± 0.1539 | 0.6960 ± 0.1387 | 0.6794 ± 0.1264 |
| | In Conf. (Carlini et al., 2019a) | 0.7254 ± 0.0214 | 0.6528 ± 0.0042 | 0.5978 ± 0.0131 | 0.5800 ± 0.0051 | 0.5669 ± 0.0106 |
| | SSFT (Maini et al., 2022) | 0.9498 ± 0.0042 | 0.9360 ± 0.0020 | 0.9077 ± 0.0023 | 0.8910 ± 0.0050 | 0.8710 ± 0.0071 |
| | Curv. (Garg et al., 2024) | 0.9800 ± 0.0003 | 0.9819 ± 0.0006 | 0.9934 ± 0.0002 | 0.9932 ± 0.0001 | 0.9932 ± 0.0006 |
| | CSL Ravikumar et al. (2025a) | **0.9845 ± 0.0026** | **0.9864 ± 0.0004** | 0.9870 ± 0.0003 | 0.9904 ± 0.0005 | 0.9906 ± 0.0003 |
| | **CSG (Ours)** | 0.9783 ± 0.0009 | 0.9809 ± 0.0011 | **0.9936 ± 0.0002** | **0.9934 ± 0.0001** | **0.9935 ± 0.0006** |
| | SGAL (Ours) | 0.9115 ± 0.0057 | 0.8835 ± 0.0128 | 0.8664 ± 0.0070 | 0.8553 ± 0.0145 | 0.8470 ± 0.0283 |
| CIFAR-100 | CL (Northcutt et al., 2021) | 0.8733 ± 0.0010 | 0.8536 ± 0.0006 | 0.7030 ± 0.1565 | 0.6833 ± 0.1427 | 0.6662 ± 0.1289 |
| | In Conf. (Carlini et al., 2019a) | 0.7069 ± 0.0069 | 0.6884 ± 0.0053 | 0.6493 ± 0.0075 | 0.6324 ± 0.0051 | 0.6257 ± 0.0044 |
| | SSFT (Maini et al., 2022) | 0.8784 ± 0.0030 | 0.8664 ± 0.0024 | 0.8358 ± 0.0008 | 0.8203 ± 0.0016 | 0.8043 ± 0.0061 |
| | Curv. (Garg et al., 2024) | 0.9876 ± 0.0021 | 0.9892 ± 0.0012 | 0.9931 ± 0.0004 | 0.9931 ± 0.0004 | 0.9932 ± 0.0002 |
| | CSL (Ravikumar et al., 2025a) | 0.9891 ± 0.0003 | 0.9895 ± 0.0002 | 0.9902 ± 0.0002 | 0.9904 ± 0.0002 | 0.9903 ± 0.0002 |
| | **CSG (Ours)** | **0.9896 ± 0.0008** | **0.9904 ± 0.0006** | **0.9934 ± 0.0003** | **0.9936 ± 0.0002** | **0.9936 ± 0.0001** |
| | SGAL (Ours) | 0.9895 ± 0.0006 | 0.9897 ± 0.0008 | 0.9856 ± 0.0008 | 0.9863 ± 0.0009 | 0.9861 ± 0.0004 |

## 5.3 EARLY STOPPING

**Experiment.** We examine how sample gradient early stopping affects calibration and privacy in a ResNet18 (He et al., 2016) on CIFAR-100. Prior work shows early stopping improves both (Ji et al., 2021): well-calibrated models make more reliable predictions, and reduced overfitting lowers privacy risk. We focus on the first descent, which offers stronger privacy with reasonable generalization. Calibration is measured with ECE, MCE, MSCE, and UCE (see metric definitions in the Appendix B.3); privacy with LiRA (Carlini et al., 2022), curvature (Ravikumar et al., 2024b), and other attacks (Sablayrolles et al., 2019; Watson et al., 2022; Yeom et al., 2018). Appendix B.3 provides additonal setup details.

**Results.** Table 2 reports calibration for all checkpoints, with reliability diagrams in Figure 6. Lower error indicates better calibration. We prioritize MCE and UCE in this context because real-world data distributions often follow a long-tail or power-law distribution. As discussed by Feldman (2020), deep learning models rely on memorizing these rare, atypical examples to achieve generalization. ECE is an expectation-based metric; it weights the calibration error of each bin by the number of samples in that bin. Consequently, in long-tailed datasets, low-accuracy bins (the tail) often contain fewer samples and are down weighted by ECE, masking miscalibration on difficult examples. In contrast, MCE and UCE treat calibration across the accuracy range more equally. When the goal is trustworthy AI that performs reliably on both head and tail data, these metrics provide a more accurate reflection of model safety. Privacy results (AUROC, balanced accuracy) are in Table 3, where smaller values imply less training data leakage. Further low-FPR MIA results are in Appendix A.1.

**Takeaways.** The sample gradient checkpoint and has lower calibration errors compared to last epoch checkpoint. Membership-inference performance drop for early-stopped model, indicating noticeably less privacy risk. In short, sample gradient early stopping reproduces the benefits of conventional validation-based early stopping without needing a separate validation set!

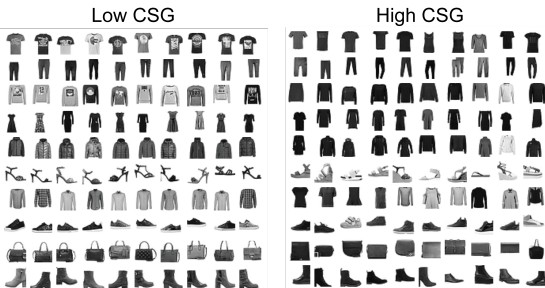

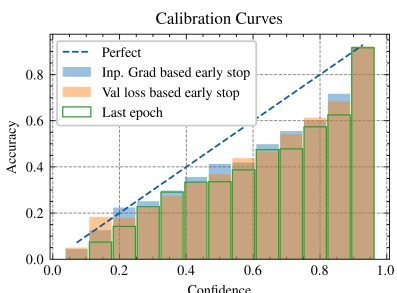

Figure 5: Using the CSG uncovers a bias in the FM-NIST dataset: darker clothing with lower contrast is often identified as high CSG (i.e. harder).

Figure 6: Reliability Diagram for Inp Grad/weight norm based, val. based early stopping and the last checkpoint.

### 5.4 MISLABEL DETECTION

**Experiment.** We detect mislabeled samples by thresholding CSG values and compare this method to state-of-the-art approaches on CIFAR-10/100 with ResNet18 under $5\% - 30\%$ symmetric label noise (labels flipped uniformly to other classes). Detection is evaluated with AUROC (higher is better). Appendix B.6 provides setup details.

**Results.** The results are presented in Table 4, showcasing the performance of our method and baselines (see Appendix B.6.1 for details on the baseline methods) on CIFAR-10 and CIFAR-100 at symmetric label noise levels of 5%, 10%, 20%, 25% and 30%.

**Takeaways.** CSG reliably detects mislabeled examples. It surpasses all baselines on CIFAR-100 and, on CIFAR-10, scales with noise matching curvature and CSL at 5–10% and surpassing them at higher levels with significantly lower cost, CSG delivers state-of-the-art performance efficiently.

### 5.5 BIAS

CSG analysis uncovers hidden biases in training data. In CIFAR-100 (Figures 1a and 1b), images of boys in red shirts or red trucks consistently receive low CSG scores, indicating they are learned earlier. This is likely due to over representation of red colored objects. This color-specific ease of learning suggests a bias that may hinder generalization. A similar trend is observed in Fashion-MNIST (Xiao et al., 2017), where Figure 5 shows that high CSG samples, learned later in training, tend to be darker and lower in contrast. These findings show that irrelevant features (e.g., color, brightness, contrast) can bias learning, and CSG offers a practical way to detect them. *Finally, to ensure reproducibility, details of all the experiments are provided in Appendix B, and the corresponding code is included in the supplementary material.*

## 6 CONCLUSION

In conclusion, we propose Cumulative Sample Gradient (CSG), a theoretically grounded and efficient proxy for memorization. CSG aligns closely with memorization (Feldman & Zhang, 2020) while being up to five orders of magnitude faster, and is $140\times$ faster than curvature (Garg et al., 2024) and $5\times$ faster than CSL (Ravikumar et al., 2025a), two prior state-of-the-art proxies. Our analysis establishes a theoretical connection between CSG, learning time, and memorization. Experimentally, we show that CSG enables validation-set-free early stopping and supports the development of an additional proxy, SGAL. Further experiments demonstrate state-of-the-art performance in mislabeled data and bias detection, positioning CSG as a scalable and interpretable tool for data-centric deep learning.

### ACKNOWLEDGMENTS

This work was supported in part by the Center for the Co-Design of Cognitive Systems (COCOSYS), a DARPA sponsored JUMP center, Semiconductor Research Corporation (SRC), National Science Foundation (NSF) award DMS-2502560, and Collins Aerospace. Efstathia Soufleri has been par-

tially supported by project MIS 5154714 of the National Recovery and Resilience Plan Greece 2.0 funded by the European Union under the NextGenerationEU Program.

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

APPENDIX

# A  ADDITIONAL BACKGROUND ON STOCHASTIC GRADIENT DESCENT (SGD)

$\rho$**-Lipschitz Gradient.** The gradient of the loss function $\ell$ is said to be $\rho$-Lipschitz on $\mathrm{Range}(\mathbf{W})$ if, for all $\vec{w}_1, \vec{w}_2 \in \mathrm{Range}(\mathbf{W})$, there exists a constant $\rho > 0$ such that:

$$\|\nabla_{w_1}\ell(\vec{w}_1) - \nabla_{w_2}\ell(\vec{w}_2)\| \le \rho\|\vec{w}_1 - \vec{w}_2\| \tag{8}$$

**Bounded Gradient Variance.** Let $\nabla_{w_t}\ell(\mathbf{w}_t)$ denote the true gradient, and let $\widetilde{\nabla}_{w_t}\ell(\mathbf{w}_t)$ be an unbiased estimator of this gradient. The estimator is said to have variance bounded by $\Gamma_v^2$ if

$$\mathbb{E}\left[\|\widetilde{\nabla}_{w_t}\ell(\vec{w}_t) - \nabla_{w_t}\ell(\vec{w}_t)\|_2^2\right] \le \Gamma_v^2 \tag{9}$$

In SGD, model parameters $\vec{w}_t$ at iteration $t$ are updated using the gradient of the loss function computed with a mini-batch or a single random sample $\vec{z}_i$. The update rule is

$$\vec{w}_{t+1} = \vec{w}_t - \eta_t\widetilde{\nabla}_{w_t}\ell(\vec{w}_t, \vec{z}_i) \tag{10}$$

Here, $\eta_t$ is the learning rate, and $\tilde{\nabla}_{w_t}\ell(\vec{w}_t, \vec{z}_i)$ is an unbiased stochastic gradient estimator.

## A.1  MEMBERSHIP INFERENCE ATTACKS (MIA)

Membership Inference Attacks (MIA) aim to determine whether a specific data point was part of a machine learning model's training dataset. By analyzing the model's outputs, such as confidence scores or loss values, attackers can infer the membership status of individual data points, posing significant privacy concerns, especially when models are trained on sensitive information (Shokri et al., 2017).

**Curvature Clues (Ravikumar et al., 2024b).** This approach leverages the curvature of the loss function with respect to input data to distinguish between training and non-training samples. Specifically, it examines the trace of the Hessian matrix (input loss curvature) of the loss function. Empirical studies have shown that this curvature tends to be higher for training data points compared to non-training ones, enabling effective membership inference even in black-box settings.

**Likelihood Ratio Attack (LiRA) (Carlini et al., 2022).** LiRA is a black-box membership inference technique that utilizes shadow models to estimate the likelihood of a data point being part of the training set. By comparing the loss distributions of shadow models trained with and without specific data points, LiRA computes a likelihood ratio to infer membership. This method has demonstrated superior performance, especially at low false positive rates, compared to previous approaches.

**Shadow Models** are auxiliary models trained to mimic the behavior of the target model. By training these models on known datasets, attackers can observe how the model behaves with known members and non-members, creating a reference for inferring membership in the target model.

In our study, we employed *64 shadow models* to capture a diverse range of behaviors, enhancing the robustness and accuracy of our membership inference attacks.

# B  EXPERIMENTAL DETAILS

## B.1  VALIDATING THEORY SETUP DETAILS

**Datasets.** We use CIFAR-100 (Krizhevsky et al., 2009) and ImageNet (Russakovsky et al., 2015) datasets. For experiments that use memorization scores, we use the pre-computed stability-based memorization scores from Feldman & Zhang (2020) which have been made publicly available by the authors.

**Architectures.** Since we use the precomputed stability-based memorization scores from Feldman & Zhang (2020) we use the same model architectures as Feldman & Zhang (2020), specifically we use Inception Szegedy et al. (2016) for CIFAR-100 and ResNet50 He et al. (2016) for ImageNet.

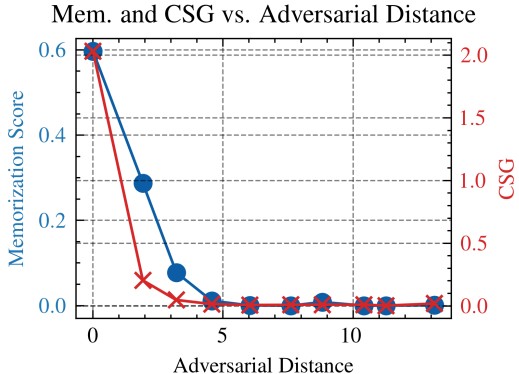

Figure 7: Comparison of CSG with adversarial distance and memorization.

**Training.** When training on CIFAR-100 the initial learning rate was set to 0.1 and trained for 200 epochs. The learning rate is decreased by 10 at epochs 120 and 160. The batch size is set to 128. We use stochastic gradient descent for training with momentum set to 0.9 and weight decay set to $1 \times 10^{-4}$. For CIFAR-100, we used the following sequence of data augmentations for training: resize ($32 \times 32$), random crop, and random horizontal flip, this is followed by normalization. For ImageNet we trained a ResNet50 with the resize random crop was set to $224 \times 224$.

## B.2 SIMILARITY WITH MEMORIZATION SETUP DETAILS

**Setup.** The datasets, architectures and training for this experiment is the same as described in section B.1.

**Similarity Score.** For cosine similarity score we use pytorch and for Pearson correlation sklearn's implementation.

**Effect of Architecture.** The table below shows the effect of using a different architecture from that of Feldman & Zhang (2020) and using the proposed CSG proxy to calculate similarity with the precomputed memorization scores from Feldman & Zhang (2020).

Table 5: CSG similarity with FZ Memorization vs Architecture. Cosine Similarity (CS) refers to the similarity with Feldman & Zhang (2020) precomputed memorization scores.

| Dataset | Architecture | Cosine Similarity (CS) |
|---------|-------------|------------------------|
| CIFAR100 | Inception (Same as FZ) | 0.84 |
| CIFAR100 | ResNet18 | 0.77 |
| ImageNet | ResNet50 (Same as FZ) | 0.78 |
| ImageNet | ResNet18 | 0.68 |

**Using MIA to test Memorization.** We use the state-of-the-art membership inference attack proposed by Ravikumar et al. (2024b) to evaluate the memorization behavior of models. Membership inference attacks measure if a given sample was used to train the model. The success of the attacks reveals privacy risks for training examples. To demonstrate that CSG captures similar properties to the memorization score introduced by Feldman and Zhang Feldman & Zhang (2020), we plot MIA AUROC against both the memorization score and CSG in Figures 9b and 9a, respectively. Both plots clearly show a monotonic increase in MIA attack success as either the memorization score or CSG increases, indicating that both metrics effectively reflect model memorization.

## B.3 EARLY STOPPING DETAILS AND SETUP

**Setup.** For this experiment we trained a ResNet18 on CIFAR-100. The initial learning rate was set to 0.1 and trained for 300 epochs. The learning rate is decreased by 10 at epochs 180 and 240. The

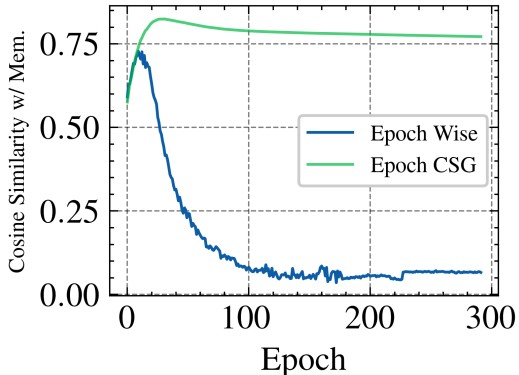

Figure 8: Epoch-wise similarity for ResNet18 on CIFAR-100 (same model and trajectory) as used in Figure 1c. We clearly observe the same trend as in Figure 3c. Figure 3c was included in the main paper because it was obtained using the FZ Inception architecture, which is the same architecture used to evaluate memorization scores in Feldman & Zhang (2020). However, this result demonstrates that our observations generalize across network architectures, provided there is sufficient parameterization.

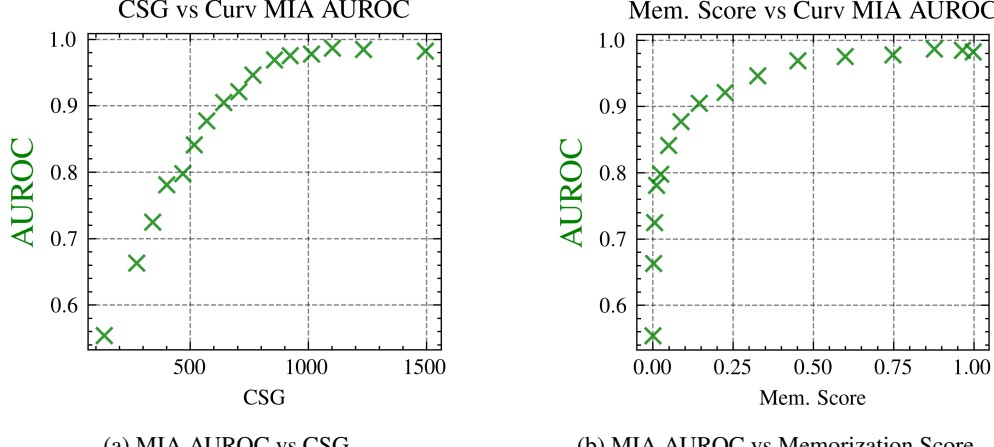

(a) MIA AUROC vs CSG.

(b) MIA AUROC vs Memorization Score.

Figure 9: Demonstrating that CSG captures similar properties as the memorization score introduced by Feldman and Zhang Feldman & Zhang (2020), we plot MIA AUROC against both the memorization score and CSG in Figures 9a and 9b, respectively. Both plots show a clear monotonic relationship: as either memorization or CSG increases, the success rate of the MIA attack also increases.

Table 6: Comparison of Cosine Similarity (CS) and Pearson Correlation (PC) of different methods with precomputed memorization scores from Feldman & Zhang (2020) Experiments are performed on the CIFAR100 dataset using the Inception model architecture (same as Feldman & Zhang (2020)) across Adam, AdamW, AdaGrad, and RMSProp optimizers. "Top 5k" refers to selecting the 5000 most memorized examples based on Feldman & Zhang (2020), and the cosine similarity and Pearson correlation for these examples is reported.

| Method | Compute Cost | Dataset | Adam | | AdamW | | AdaGrad | | RMSProp | |
|---|---|---|---|---|---|---|---|---|---|---|
| | | | CS | PC | CS | PC | CS | PC | CS | PC |
| **SGAL (Ours)** | 0.1-0.3× | All | 0.87 | 0.79 | 0.87 | 0.78 | 0.85 | 0.74 | 0.87 | 0.78 |
| CSL (Ravikumar et al., 2025b) | 1× | All | 0.87 | 0.78 | 0.85 | 0.75 | 0.84 | 0.73 | 0.87 | 0.78 |
| Loss Sensitivity (Arpit et al., 2017) | 1× | All | 0.85 | 0.77 | 0.84 | 0.75 | 0.82 | 0.70 | 0.85 | 0.77 |
| **CSG (Ours)** | 0.1-0.3× | All | 0.84 | 0.74 | 0.84 | 0.72 | 0.82 | 0.69 | 0.84 | 0.73 |
| Forgetting Frequency (Toneva et al., 2019) | 1× | All | 0.83 | 0.72 | 0.79 | 0.63 | 0.74 | 0.54 | 0.83 | 0.71 |
| VoG All Ckpts (Agarwal et al., 2022) | 1× | All | 0.80 | 0.66 | 0.78 | 0.62 | 0.80 | 0.65 | 0.81 | 0.67 |
| Final Sample Loss | 1× | All | 0.09 | 0.06 | 0.04 | 0.03 | 0.45 | 0.31 | 0.09 | 0.06 |
| VoG Last 5 Ckpts (Agarwal et al., 2022) | 1× | All | 0.22 | 0.14 | 0.12 | 0.08 | 0.47 | 0.31 | 0.25 | 0.17 |
| TracIn (Pruthi et al., 2020) | 26× (measured) | All | 0.85 | 0.75 | 0.83 | 0.70 | 0.85 | 0.75 | 0.85 | 0.75 |
| GraNd (Paul et al., 2021) | 7.8× (measured) | All | 0.81 | 0.69 | 0.79 | 0.65 | 0.81 | 0.69 | 0.69 | 0.39 |
| Loss Sensitivity (Arpit et al., 2017) | 1× | Top 5k | 0.79 | 0.69 | 0.78 | 0.67 | 0.75 | 0.67 | 0.79 | 0.69 |
| **SGAL (Ours)** | 0.1-0.3× | Top 5k | 0.83 | 0.73 | 0.82 | 0.72 | 0.80 | 0.67 | 0.83 | 0.73 |
| CSL (Ravikumar et al., 2025b) | 1× | Top 5k | 0.83 | 0.73 | 0.81 | 0.69 | 0.80 | 0.67 | 0.83 | 0.73 |
| **CSG (Ours)** | 0.1-0.3× | Top 5k | 0.78 | 0.64 | 0.72 | 0.56 | 0.71 | 0.54 | 0.78 | 0.64 |
| Final Sample Loss | 1× | Top 5k | 0.08 | 0.05 | 0.04 | 0.03 | 0.40 | 0.27 | 0.08 | 0.05 |
| Forgetting Frequency (Toneva et al., 2019) | 1× | Top 5k | 0.79 | 0.67 | 0.93 | 0.05 | 0.89 | 0.07 | 0.79 | 0.66 |
| VoG All Ckpts (Agarwal et al., 2022) | 1× | Top 5k | 0.75 | 0.60 | 0.73 | 0.55 | 0.74 | 0.57 | 0.76 | 0.60 |
| VoG Last 5 Ckpts (Agarwal et al., 2022) | 1× | Top 5k | 0.19 | 0.12 | 0.11 | 0.08 | 0.42 | 0.27 | 0.22 | 0.15 |
| TracIn (Pruthi et al., 2020) | 26× (measured) | Top 5k | 0.80 | 0.67 | 0.77 | 0.63 | 0.80 | 0.62 | 0.80 | 0.68 |
| GraNd (Paul et al., 2021) | 7.8× (measured) | Top 5k | 0.76 | 0.62 | 0.74 | 0.59 | 0.76 | 0.62 | 0.63 | 0.32 |

batch size is set to 128. We use stochastic gradient descent for training with momentum set to 0.9 and weight decay set to $1 \times 10^{-4}$.

**Model calibration** refers to the extent to which a model's predicted probabilities reflect the true likelihood of outcomes. A well-calibrated model not only makes accurate predictions but also assigns confidence scores that correspond closely to the observed frequencies of correct classifications. For example, among all predictions assigned a confidence of 70%, approximately 70% should be correct. Proper calibration is crucial in applications where decision-making depends on reliable uncertainty estimates, such as medical diagnosis or autonomous driving.

To evaluate calibration, *reliability diagrams* (also known as calibration plots) are widely used. These plots compare predicted confidence with empirical accuracy by partitioning predictions into $M$ bins based on their confidence scores (e.g., intervals like [0.0, 0.1), [0.1, 0.2), ..., [0.9, 1.0]). For each bin $B_m$, the accuracy $\mathrm{acc}(B_m)$ is the fraction of correct predictions, and the confidence $\mathrm{conf}(B_m)$ is the average predicted probability. Discrepancies between these two values indicate miscalibration.

Several quantitative metrics are used to summarize calibration performance, see Wang (2023) for a detailed analysis:

$$\text{Expected Calibration Error (ECE)} = \sum_{m=1}^{M} \frac{|B_m|}{n} \left| \mathrm{acc}(B_m) - \mathrm{conf}(B_m) \right|, \quad (11)$$

$$\text{Maximum Calibration Error (MCE)} = \max_{m=1,...,M} \left| \mathrm{acc}(B_m) - \mathrm{conf}(B_m) \right|, \quad (12)$$

$$\text{Mean Squared Calibration Error (MSCE)} = \frac{1}{M} \sum_{m=1}^{M} \left( \mathrm{acc}(B_m) - \mathrm{conf}(B_m) \right)^2, \quad (13)$$

$$\text{Uniform Calibration Error (UCE)} = \frac{1}{M} \sum_{m=1}^{M} \left| \mathrm{acc}(B_m) - \mathrm{conf}(B_m) \right|. \quad (14)$$

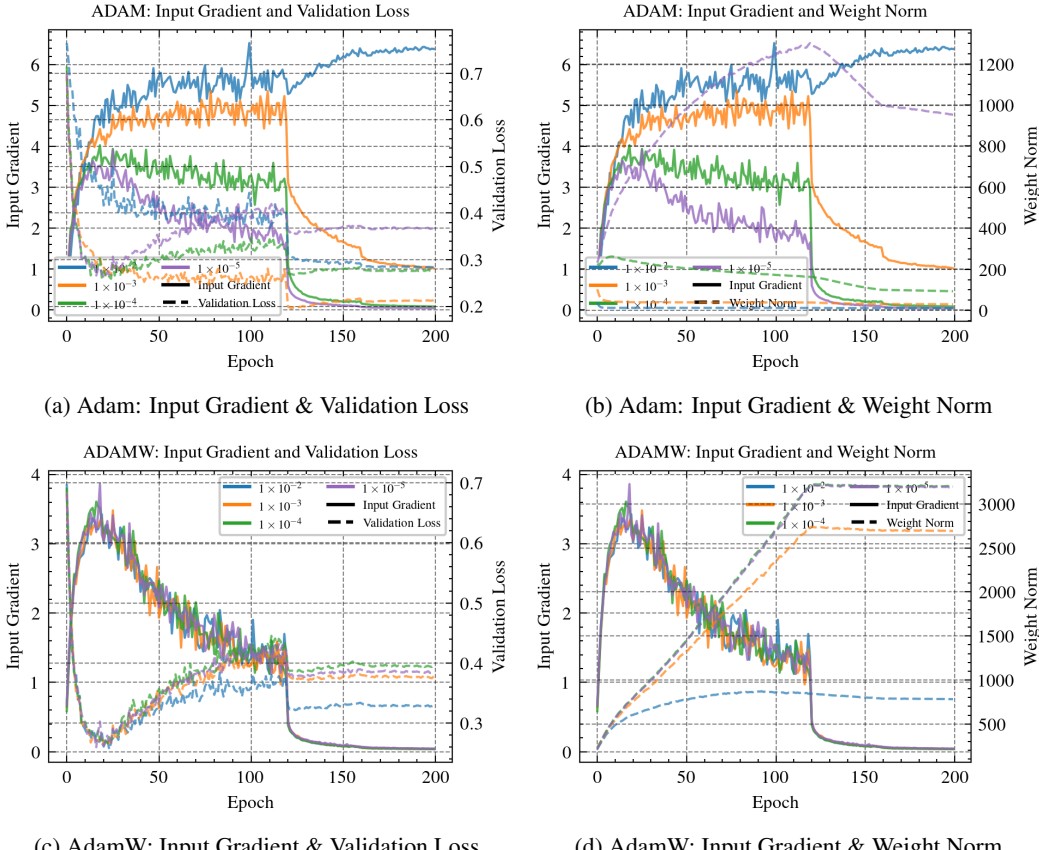

Figure 10: Comparison of input gradients against Validation Loss (Left column) and Weight Norm (Right column) for Adam and AdamW.

ECE computes the weighted average of absolute errors across all bins, giving more influence to bins with more samples. MCE captures the worst-case miscalibration observed across bins. MSCE penalizes larger discrepancies more severely by squaring the calibration errors. UCE provides an unweighted average of absolute differences, treating all bins equally regardless of sample size. Together, these metrics offer a robust framework for assessing and improving the reliability of probabilistic predictions.

where

$$\mathrm{acc}(B_m) = \frac{1}{|B_m|} \sum_{i \in B_m} \mathbf{1}\{\hat{y}_i = y_i\}, \quad \mathrm{conf}(B_m) = \frac{1}{|B_m|} \sum_{i \in B_m} \hat{p}_i, \quad (15)$$

and $n$ is the total number of samples, $M$ the number of bins, and $B_m$ the set of indices assigned to bin $m$.

**MIA Setup.** For the MIA test setup we use the code, models and setup from Ravikumar et al. (2024b).

## B.4 ADDITIONAL RESULTS WITH ADAM OPTIMIZER

We ran additional experiments with the Adam optimizer. For the training procedure, a ResNet18 model was trained on CIFAR100 using Adam with an initial learning rate of $1 \times 10^{-3}$ and a weight decay of $1 \times 10^{-4}$. We employed the standard `CrossEntropyLoss` function and a `MultiStepLR` scheduler, which decayed the learning rate by a factor of 0.1 at 60% and 80% of the 200 total training epochs.

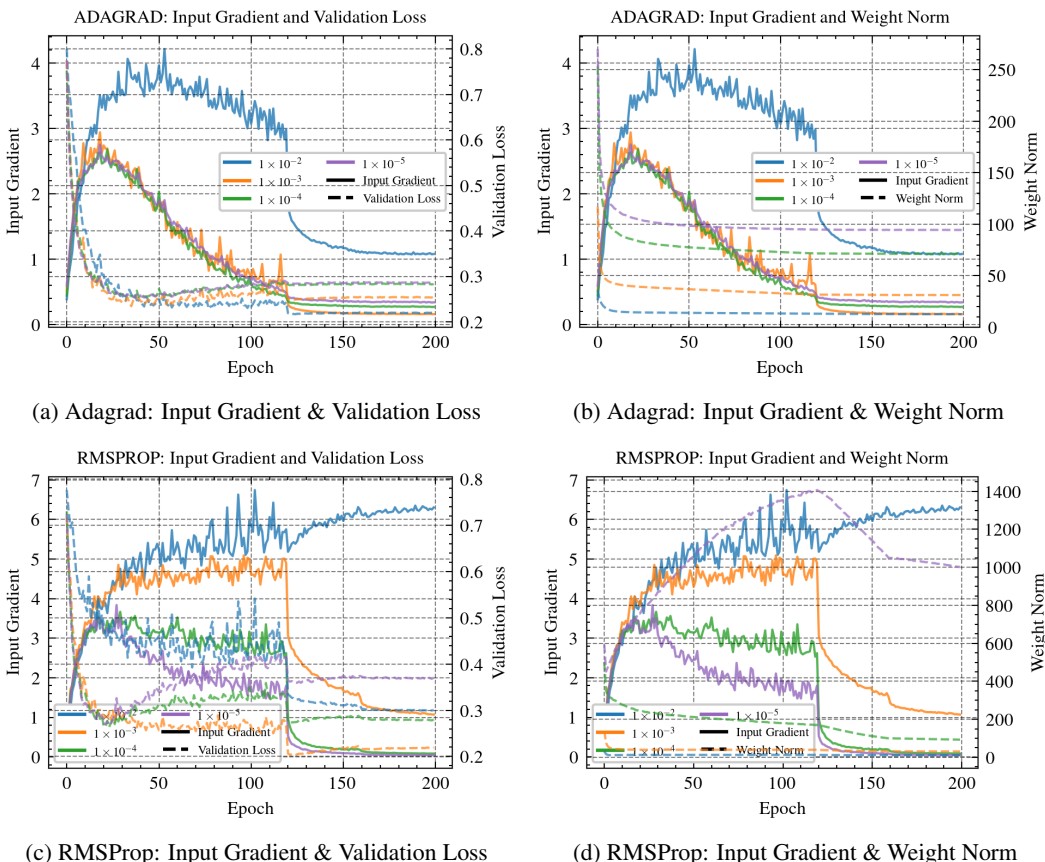

(a) Adagrad: Input Gradient & Validation Loss

(b) Adagrad: Input Gradient & Weight Norm

(c) RMSProp: Input Gradient & Validation Loss

(d) RMSProp: Input Gradient & Weight Norm

Figure 11: Comparison of input gradients against Validation Loss (Left column) and Weight Norm (Right column) for Adagrad, and RMSProp.

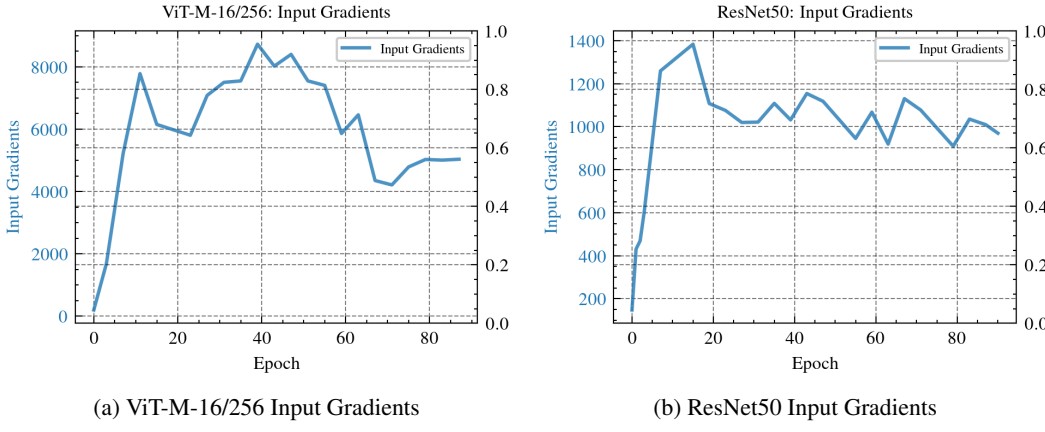

(a) ViT-M-16/256 Input Gradients

(b) ResNet50 Input Gradients

Figure 12: Evolution of input gradients during training on ImageNet. Subfigure (a) shows the gradients for ViT-M-16/256, while (b) shows ResNet50.

The results are reported in Table 7. These results show a gradient peaking phenomenon similar to that observed with SGD. This potentially hints at a more general behavior of neural networks that may be optimizer-agnostic.

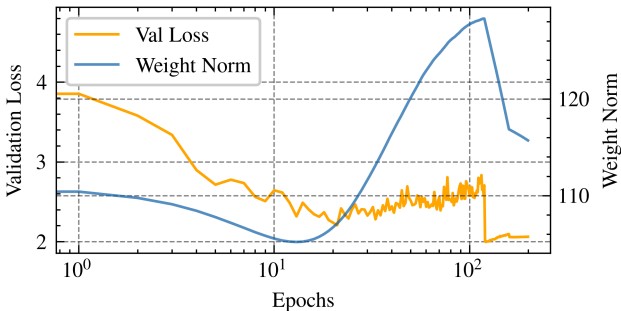

Figure 13: Weight norm behavior on ImageNet using a modified version of the standard recipe for training a ResNet18. The recipe trained longer for 200 epochs (instead of 90) with larger batch size and weight decay. The plot shows an initial decay, followed by an increase and a subsequent decrease. The first validation loss minimum roughly coincides with the first weight norm minimum (of by a few epochs but exaggerated here due to the log scale plot and the points are obtained once per epoch).

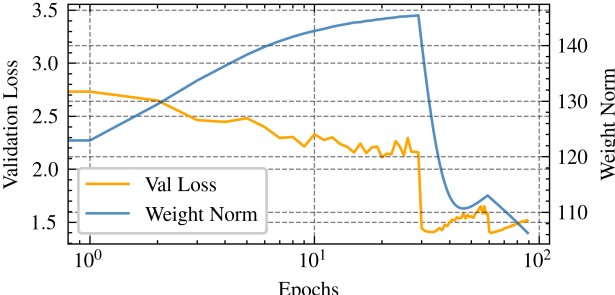

Figure 14: Weight norm behavior on ImageNet using a standard recipe for training a ResNet18. The recipe trained for 90 epochs with batch size 128 weight decay 1e-4, with SGD momentum of 0.9. The plot shows the same peaky behavior as CIFAR100 from Figure 3b, where the peak of the weight norm coincides with the minimum validation loss for the first descent.

Table 7: Gradient norm values for ResNet18 trained on CIFAR100 using Adam optimizer.

| Epoch | Input Grad Norm |
|-------|-----------------|
| 1     | 1.2486          |
| 10    | 3.4195          |
| 50    | 3.1032          |
| 80    | 3.1794          |
| 100   | 3.6039          |
| 120   | 1.0826          |
| 150   | 0.2104          |
| 180   | 0.0595          |

## B.5 COMPUTE COST ANALYSIS

In this section, we evaluate the computational overhead associated with our proposed proxy in comparison to existing approaches. Let us define the cost of a single forward pass through a neural network as $F$. A standard backpropagation step, involving both forward and backward passes, therefore incurs a total cost of $3F$. With this in mind, and denoting the dataset size by $m$ and the number of training epochs by $T$, we outline the cost incurred by different methods below.

**Stability-Based Memorization.** As introduced by Feldman & Zhang (2020), this approach involves training between 2,000 and 10,000 models to derive memorization scores. Consequently, the computational burden scales with the number of models trained, resulting in a total cost of approximately:

$$10{,}000 \cdot 3F \cdot T \cdot m$$

**Cumulative Sample Curvature.** The authors of Garg et al. (2024) propose estimating memorization through average sample curvature computed throughout training. This involves applying Hutchinson's trace estimator, which, for each sample per epoch, requires two forward passes and one backward pass, repeated $n$ times. For a favorable comparison, we assume $n = 2$, although better empirical performance is achieved with $n = 10$. The resulting cost combines standard training and curvature estimation:

$$\text{Cost} = 3F \cdot T \cdot m + 4F \cdot T \cdot m \cdot n \tag{16}$$
$$= 3F \cdot T \cdot m + 4F \cdot T \cdot m \cdot 2 \tag{17}$$
$$= 11F \cdot T \cdot m \tag{18}$$

If using $n = 10$ to match the best results reported in Tables 4 the cost increases to:

$$43F \cdot T \cdot m$$

**CSG (Ours).** Our proposed method, CSG, is designed to introduce minimal additional computational overhead. It utilizes gradient information that is already computed during training and only needs to be evaluated during the initial phase—typically between 10% and 30% of total training iterations. While computing input gradients involves an extra matrix multiplication (multiplying the error from backpropagation by the corresponding weight matrix), this cost is negligible relative to the overall training workload. Thus, the total cost is a fraction of standard training:

$$\text{Cost} = 0.1 \cdot 3F \cdot T \cdot m = 0.3F \cdot T \cdot m$$

A comparative summary of computational costs is provided in Table 8.

## B.6 MISLABELED DETECTION SETUP DETAILS

**Area Under the Receiver Operating Characteristic** (AUROC or simply AUC-ROC) is a performance metric used to evaluate the discriminative ability of a binary classification model. It summarizes the relationship between the True Positive Rate (TPR) and the False Positive Rate (FPR) across all possible classification thresholds. TPR, also known as sensitivity or recall, and FPR are defined

Table 8: Summary of the compute cost of the proposed metric compared to existing methods.

| Method | Absolute Cost | Relative Cost |
|---|---|---|
| Stability-Based (Feldman & Zhang, 2020) | $6000FTm - 30,000FTm$ | $2,000 \times - 30,000 \times$ |
| Input Curvature (Garg et al., 2024) | $11FTm - 43FTm$ | $3.6 \times - 14.33 \times$ |
| **CSG (Ours)** | $0.3FTm$ | $0.1 \times - 0.3 \times$ |

as:

$$\text{TPR} = \frac{\text{TP}}{\text{TP} + \text{FN}}, \tag{19}$$

$$\text{FPR} = \frac{\text{FP}}{\text{FP} + \text{TN}}, \tag{20}$$

where TP, FN, FP, and TN refer to true positives, false negatives, false positives, and true negatives, respectively. The ROC curve is generated by plotting TPR against FPR as the classification threshold is varied from 0 to 1. The area under this curve (AUROC) provides a single scalar value representing the model's ability to distinguish between the two classes. An AUROC of 1.0 indicates perfect class separation, while a value of 0.5 suggests the model performs no better than random guessing. Low AUROC values close to 0.5 imply poor model performance, whereas values closer to 1.0 indicate strong predictive capability.

**Setup.** Setup. We trained a ResNet-18 model on the CIFAR-10 and CIFAR-100 datasets. The initial learning rate was set to 0.1, and training was conducted for 300 epochs. The learning rate was reduced by a factor of 10 at epochs 180 and 240. A batch size of 128 was used. Training was performed using stochastic gradient descent (SGD) with a momentum of 0.9 and a weight decay of $1 \times 10^{-4}$. The CSG circulations spanned the entire 300 training epochs, yielding a speedup of approximately 14× over curvature-based methods. To introduce label noise, a specified percentage of the training set was randomly selected and had their labels reassigned to incorrect classes. For instance, in the 5% noise setting, 5% of the 50,000 training examples were randomly chosen and their labels were flipped to another classes. No validation split was used; the model was trained on the full noisy training set, and performance metrics were evaluated on the standard test set.

### B.6.1 BASELINE METHODS

Next, we elaborate on the experimental setup used for identifying mislabeled data points.

**Threshold Based Learning Time.** This refers to the first epoch during which a training sample is correctly classified. For instance, if the prediction sequence over epochs is $[0, 0, 0, 1, 1, 0, 1, 1, 0, 1, 1, 1]$, the learning time is recorded as epoch 3, the first occurrence of a correct prediction.

**In Confidence.** As defined in Carlini et al. (2019a), the in-confidence score is computed as $1 - p$, where $p$ is the predicted probability assigned to the true label.

**Confident Learning.** We implemented confident learning following the approach of Northcutt et al. (2021), using the cleanlab package (https://github.com/cleanlab/cleanlab). To ensure reliable predictions, we employed 3-fold cross-validation to estimate out-of-sample class probabilities, which were then used with cleanlab to identify likely label errors.

**Second Split Forgetting Time (SSFT).** (Maini et al., 2022) is computed by training the model sequentially on two disjoint subsets, Set 1 followed by Set 2. The forgetting time corresponds to how quickly samples from the first set become misclassified during fine-tuning on the second. This procedure is reversed to cover the complete dataset, with forgetting times tracked symmetrically for both splits.

**Curvature.** To measure the curvature of individual training samples, we followed the method proposed in Garg et al. (2024), using hyperparameters $n = 10$ and $h = 0.001$, in line with their experimental setup.

**Compute Cost.** Compared to alternative methods, some approaches introduce considerable computational overhead. For instance, Confidence Learning (CL) involves training multiple models using

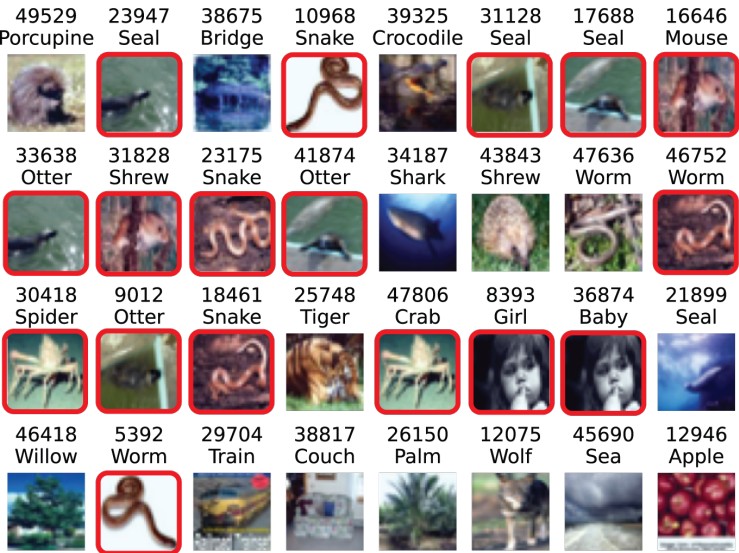

Figure 15: Visualization of the 32 highest CSG-scoring images from CIFAR-100, annotated with their class labels and PyTorch image indices. Notably, image indices 23947 and 33638 appear to be identical images but are assigned different labels. All such duplicate pairs in the figure are highlighted. Additional pairs were discovered using this technique, including one for image index 31487 (not shown here), which is labeled as a 'whale'.

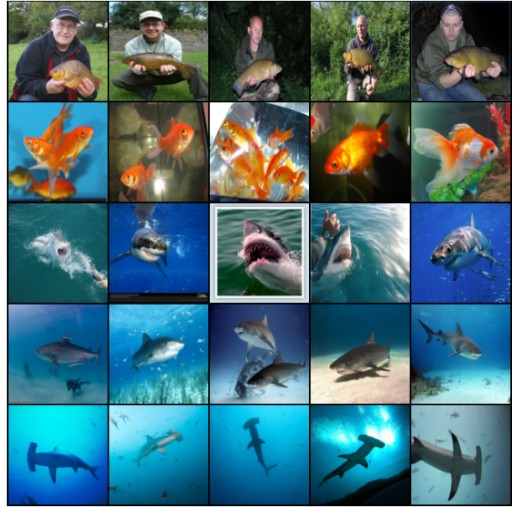
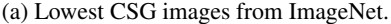

(a) Lowest CSG images from ImageNet.

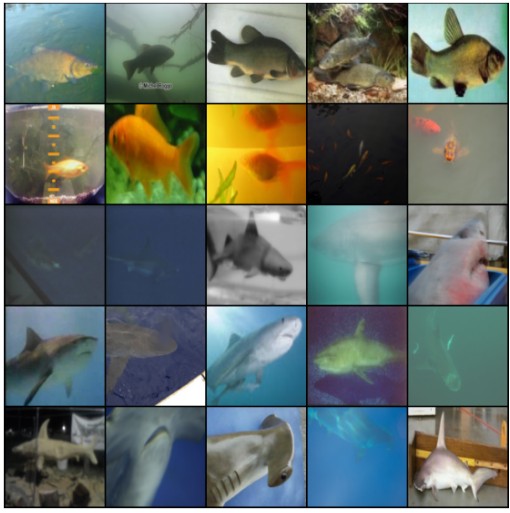

(b) Highest CSG images from ImageNet.

Figure 16: (a) Lowest CSG images from ImageNet (first few classes). Captures examples prototypical for the class i.e. easy examples. (b) Highest CSG images from ImageNet (for the same first few classes). High CSG captures examples atypical for the class i.e. hard examples, likely memorized.

a k-fold strategy (three in this case) which substantially raises the computational burden. Similarly, SSFT necessitates training on at least two distinct subsets of the dataset, effectively doubling the training effort compared to a standard setup. Other techniques, such as those relying on input loss curvature, are even more resource-intensive, demanding over an order of magnitude more computational power. CSG offers comparable training costs to vanilla training with minimal overhead (one extra matrix multiplication). Overall, this highlights the trade-offs between computational efficiency and performance when selecting among existing methods.

## B.7 MIA Experiments

**Additional Results** We present the results of membership inference attack (MIA) detection on early-stopped models, using our proposed technique, validation loss, and the final epoch. These results supplement the main paper by providing true positive rates (TPR) at very low false positive rates (FPR), complementing the previously reported AUROC and balanced accuracy metrics.

Table 9: Mean $\pm$ standard deviation of True Positive Rates (TPR in %) at different False Positive Rates (FPR) for various methods and stopping points.

| Method | Ours | | | Val. Loss | | | Last Epoch | |
|---|---|---|---|---|---|---|---|---|
| FPR | 1e-4 | 1e-3 | 1e-2 | 1e-4 | 1e-3 | 1e-2 | 1e-4 | 1e-3 | 1e-2 |
| Curvature Ravikumar et al. (2024b) | $0.34 \pm 0.35$ | $1.54 \pm 1.79$ | $4.51 \pm 3.39$ | $0.22 \pm 0.17$ | $1.97 \pm 1.90$ | $5.03 \pm 2.67$ | $15.94 \pm 0.28$ | $18.99 \pm 0.22$ | $25.88 \pm 1.21$ |
| LiRA Carlini et al. (2022) | $0.03 \pm 0.01$ | $0.58 \pm 0.62$ | $1.69 \pm 0.77$ | $0.03 \pm 0.04$ | $0.43 \pm 0.40$ | $2.10 \pm 1.17$ | $0.30 \pm 0.05$ | $1.00 \pm 0.25$ | $22.77 \pm 0.36$ |
| Yeom et al. Yeom et al. (2018) | $0.01 \pm 0.01$ | $0.10 \pm 0.03$ | $1.15 \pm 0.08$ | $0.01 \pm 0.01$ | $0.10 \pm 0.02$ | $1.10 \pm 0.05$ | $0.01 \pm 0.00$ | $0.15 \pm 0.01$ | $1.42 \pm 0.08$ |
| MAST Sablayrolles et al. (2019) | $0.64 \pm 0.75$ | $1.94 \pm 1.99$ | $7.78 \pm 4.85$ | $0.54 \pm 0.39$ | $2.08 \pm 1.65$ | $8.88 \pm 4.07$ | $3.18 \pm 1.44$ | $8.22 \pm 1.27$ | $18.24 \pm 0.45$ |
| Cal. Loss Watson et al. (2022) | $0.34 \pm 0.45$ | $1.32 \pm 1.36$ | $6.18 \pm 4.02$ | $0.49 \pm 0.55$ | $1.60 \pm 1.29$ | $7.02 \pm 3.54$ | $2.11 \pm 0.72$ | $5.73 \pm 0.76$ | $13.63 \pm 0.29$ |

# C Proofs

## C.1 Proof for Lemma 4.1

The proof for this lemma build on the result from Ravikumar et al. (2025a), which states the following.

**Lemma** *For any neural network without a skip connection at the first layer, and a given mini-batch of inputs $Z_b = (X_b, Y_b)$, the Frobenius norm of the gradient of the loss $\ell$ with respect to the input is bounded by the norm of the gradient with respect to the network's weights $\vec{w}_t$. Specifically:*

$$\|\nabla_{X_b}\ell(\vec{w}_t, Z_b)\|_F \leq k_g \|\nabla_{w_t}\ell(\vec{w}_t, Z_b)\|_F \tag{21}$$

*where $k_g = \dfrac{\|W_t^{(1)}\|_F \, \|(X_b^\top)^+\|_F}{s_P}$ and $s_P$ denotes the smallest singular value of $P = X_b^\top (X_b^\top)^+$, where $^+$ denotes pseudo-inverse.*

Ravikumar et al. (2025a) provide the proof for the above result. This forms the basis of the general result. To obtain the form stated in Lemma 4.1, consider a single-sample mini-batch, in which case $\vec{x}_i$ is a column vector. In this case, the norm of the pseudoinverse is the reciprocal of its norm. Since $\vec{x}_i$ is a column vector, the smallest singular value is 1.

To see this, Let $X \in \mathbb{R}^{n \times 1}$ be a column vector. Then $X^\top \in \mathbb{R}^{1 \times n}$ is a row vector, whose Moore–Penrose pseudoinverse is given by $(X^\top)^+ = \dfrac{X}{\|X\|^2} \in \mathbb{R}^{n \times 1}$. Therefore,

$$P = X^\top (X^\top)^+ = X^\top \frac{X}{\|X\|^2} = \frac{X^\top X}{\|X\|^2} = \frac{\|X\|^2}{\|X\|^2} = 1.$$

Since $P$ is the $1 \times 1$ matrix [1], its only singular value is 1, and thus the smallest singular value of $P$ is also 1. Substituting these results into the above lemma, i.e. $Z_b = \vec{z}_i = (\vec{x}_i, y)$, $X_b = \vec{x}_i$, $\|(X_b^\top)^+\|_F = 1/\|\vec{x}_i\|_F$, we obtain the form of Lemma 4.1:

$$\|\nabla_{x_i}\ell(\vec{w}_t, \vec{z}_i)\|_F \leq \|\vec{w}_t\|_F \, \|\nabla_{w_t}\ell(\vec{w}_t, \vec{z}_i)\|_F \, / \, \|\vec{x}_i\|_F \quad \blacksquare$$

## C.2 Proof of Theorem 4.2

**Proof of Theorem 4.2**

**Assumptions.** Throughout this proof we assume:

- The loss function $\ell$ is $\rho$-Lipschitz smooth.
- The stochastic gradient estimator is unbiased: $\mathbb{E}_t[\widetilde{\nabla}_w\ell(\vec{w}_t)] = \nabla_w\ell(\vec{w}_t)$.
- The gradient variance is bounded: $\mathbb{E}\left[\|\widetilde{\nabla}_{w_t}\ell(\vec{w}_t) - \nabla_{w_t}\ell(\vec{w}_t)\|_2^2\right] \leq \Gamma_v^2$.

- The gradient noise is uncorrelated with the true gradient: $\mathbb{E}_t[\langle \nabla_{w_t} \ell(\vec{w}_t), \delta_t \rangle] = 0$.
- Assume $\beta$-stability of SGD.
- Assume $L$-bounded loss.

From the input gradient convergence result from Ravikumar et al. (2025a) (Eq. 57), for any mini-batch $X_b$ we have

$$\mathbb{E}_R\big[\|\nabla_{X_b}\ell(\vec{w}_R)\|_2^2\big] \leq \frac{\kappa_m}{T\eta}\Big(\mathbb{E}_t[\ell(\vec{w}_0) - \ell(\vec{w}_T)]\Big) + \frac{\kappa_m\eta_s\rho\Gamma_v^2}{2}. \tag{22}$$

Please note, $T$ represents some iteration step, while $R < T$ is a discrete random variable, sampled non-uniformly from the set $\{1, 2, \cdots, T\}$ (i.e. support of size $T$). This distinction is a key aspect of the proof strategy, adapted from the convergence analysis of SGD (randomized stochastic gradient) from Ghadimi & Lan (2013). The proof establishes a guarantee on the expected performance of a non-uniformly sampled iterate $R$, rather than the final iterate $T$.

Consider the case where the mini-batch $X_b$ to contains only a single training example $z_i$. This choice removes dependence on the SGD batch size (though it affects $\kappa_m$, but is still a constant), yielding

$$\mathbb{E}_R\big[\|\nabla_{x_i}\ell(\vec{w}_R)\|_2^2\big] \leq \frac{\kappa_m}{T\eta}\Big(\mathbb{E}_t[\ell(\vec{w}_0) - \ell(\vec{w}_T)]\Big) + \frac{\kappa_m\eta_s\rho\Gamma_v^2}{2}. \tag{23}$$

Let $\vec{w}_*$ represent the global optima, replacing $\ell(\vec{w}_T)$ by the optimal value $\ell(\vec{w}_*)$, since by definition $\ell(\vec{w}_*) < \ell(\vec{w}_T)$ gives the more interpretable bound

$$\mathbb{E}_R\big[\|\nabla_{x_i}\ell(\vec{w}_R)\|_2^2\big] \leq \frac{\kappa_m}{T\eta}\Big(\mathbb{E}_t[\ell(\vec{w}_0) - \ell(\vec{w}_*)]\Big) + \frac{\kappa_m\eta_s\rho\Gamma_v^2}{2}. \tag{24}$$

Observe that the left-hand side of Eq. 24 is exactly the term appearing in the learning condition. If we denote the worst-case bound by $\tau$, then for some $T$ we have

$$\tau = \frac{\kappa_m}{T\eta}\Big(\mathbb{E}_t[\ell(\vec{w}_0) - \ell(\vec{w}_*)]\Big) + \frac{\kappa_m\eta_s\rho\Gamma_v^2}{2}. \tag{25}$$

For a sample $\vec{z}_i$, this $T = T_{z_i}$. Rearranging yields

$$T_{z_i}\left[\frac{\eta\tau}{\kappa_m} - \frac{\kappa_m\eta_s\rho\Gamma_v^2}{2}\right] = \mathbb{E}_t[\ell(\vec{w}_0) - \ell(\vec{w}_*)]. \tag{26}$$

We expand the right-hand side to separate the memorization component:

$$T_{z_i}\left[\frac{\eta\tau}{\kappa_m} - \frac{\kappa_m\eta_s\rho\Gamma_v^2}{2}\right] = \mathbb{E}_{t,z_i}\big[\ell(\vec{w}_0, z_i) - \ell^{\backslash z_i}(\vec{w}_*, z_i)\big] + \mathbb{E}_{t,z_i}\big[\ell^{\backslash z_i}(\vec{w}_*, z_i) - \ell(\vec{w}_*, z_i)\big]. \tag{27}$$

The second term is exactly the memorization quantity $\text{mem}(z_i)$, scaled by $L$, see Ravikumar et al. (2024a). Taking expectation with respect to the sampling distribution $p$ gives

$$\mathbb{E}_p[T_{z_i}]\left[\frac{\eta\tau}{\kappa_m} - \frac{\kappa_m\eta_s\rho\Gamma_v^2}{2}\right] = \mathbb{E}_{t,z_i,p}\big[\ell(\vec{w}_0, z_i) - \ell^{\backslash z_i}(\vec{w}_*, z_i)\big] + \mathbb{E}_{t,z_i}[L\,\text{mem}(z_i)]. \tag{28}$$

Using the definition of $\beta$-stability (Eq. 2), yields

$$\mathbb{E}_p[T_{z_i}]\left[\frac{\eta\tau}{\kappa_m} - \frac{\kappa_m\eta_s\rho\Gamma_v^2}{2}\right] \geq \mathbb{E}_{t,p}[\ell(\vec{w}_0) - \ell(\vec{w}_*)] + \mathbb{E}_{t,z_i}[L\,\text{mem}(z_i)] - \beta. \tag{29}$$

Now consider Eq. 23, let $T = T_{max}$, we get

$$\mathbb{E}_R\left[\|\nabla_{x_i}\ell(\vec{w}_R)\|_2^2\right] \leq \frac{\kappa_m}{T_{max}\eta}\big(\mathbb{E}_t[\ell(\vec{w}_0) - \ell(\vec{w}_*)]\big) + \frac{\kappa_m\eta_s\rho\Gamma_v^2}{2} \tag{30}$$

$$\frac{\eta}{\kappa_m}\mathbb{E}_{z_i}[\text{CSG}(z_i)] \leq \mathbb{E}_t[\ell(\vec{w}_0) - \ell(\vec{w}_*)] + \frac{\kappa_m\eta_s T_{max}\rho\Gamma_v^2}{2} \tag{31}$$

$$\frac{\eta}{\kappa_m}\mathbb{E}_{z_i}[\text{CSG}(z_i)] - \frac{\kappa_m\eta_s T_{max}\rho\Gamma_v^2}{2} \leq \mathbb{E}_t[\ell(\vec{w}_0) - \ell(\vec{w}_*)] \tag{32}$$

Combining Eq. 29 this with the inequality above yields

$$\mathbb{E}_p[T_{z_i}]\left[\frac{\eta\tau}{\kappa_m} - \frac{\kappa_m\eta_s\rho\Gamma_v^2}{2}\right] \geq \mathbb{E}_{t,z_i}[L\operatorname{mem}(z_i)] + \frac{\eta}{\kappa_m}\mathbb{E}_{p,z_i}[\mathrm{CSG}(z_i)] - \frac{\kappa_m\eta_s T_{\max}\rho\Gamma_v^2}{2} - \beta$$

(33)

Finally, noting that $\operatorname{mem}(z_i) \geq 0$, and set

$$C_1 = \left[\tau - \frac{\kappa_m^2\eta_s\rho\Gamma_v^2}{2\eta}\right], \quad C_2 = \frac{\kappa_m\beta}{\eta} + \frac{\kappa_m^2\eta_s T_{\max}\rho\Gamma_v^2}{2\eta} \geq 0,$$

Since the choice of learning threshold $\tau$ is arbitrary, we can choose

$$\tau > \frac{\kappa_m^2\eta_s\rho\Gamma_v^2}{2\eta}$$

Thus $C_1 > 0$. All constants in $C_2$ are positive thus $C_2 > 0$. Rearranging the terms, we conclude

$$\boxed{\mathbb{E}_{p,z_i}[\mathrm{CSG}(\vec{z}_i)] \leq C_1\,\mathbb{E}_{p,z_i}[T_{z_i}] + C_2} \quad \blacksquare$$

(34)

### C.3 PROOF OF THEOREM 4.3

**Assumptions.** We assume:

- The loss function $\ell$ is $\rho$-Lipschitz smooth.
- The stochastic gradient estimator is unbiased: $\mathbb{E}_t[\widetilde{\nabla}_w\ell(\vec{w}_t)] = \nabla_w\ell(\vec{w}_t)$.
- The gradient variance is bounded: $\mathbb{E}\left[\|\widetilde{\nabla}_{w_t}\ell(\vec{w}_t) - \nabla_{w_t}\ell(\vec{w}_t)\|_2^2\right] \leq \Gamma_v^2$.
- The gradient noise is uncorrelated with the true gradient: $\mathbb{E}_t[\langle\nabla_{w_t}\ell(\vec{w}_t), \delta_t\rangle] = 0$. (Same as Assumption 1 from Ghadimi & Lan (2013).
- Assume $\beta$-stability of SGD.
- Assume $L$-bounded loss.

From Equation 24, we have

$$\mathbb{E}_R\left[\|\nabla_{x_i}\ell(\vec{w}_R)\|_2^2\right] \leq \frac{\kappa_m}{T\eta}\big(\ell(\vec{w}_0) - \ell(\vec{w}_*)\big) + \frac{\kappa_m\eta_s\rho\Gamma_v^2}{2}.$$

(35)

Please note, $T$ represents some iteration step, while $R < T$ is a discrete random variable, sampled non-uniformly from the set $\{1, 2, \cdots, T\}$ (i.e. support of size $T$). This distinction is a key aspect of the proof strategy, adapted from the convergence analysis of SGD (randomized stochastic gradient) from Ghadimi & Lan (2013). The proof establishes a guarantee on the expected performance of a non-uniformly sampled iterate $R$, rather than the final iterate $T$. Multiplying by $T_{z_i}$, gives

$$T_{z_i}\mathbb{E}_R\left[\|\nabla_{x_i}\ell(\vec{w}_R)\|_2^2\right] \leq \frac{\kappa_m}{\eta}\big(\ell(\vec{w}_0) - \ell(\vec{w}_*)\big) + \frac{\kappa_m\eta_s\rho\Gamma_v^2 T_{z_i}}{2}.$$

(36)

Using the definition of CSG, (Eq 3), we have,

$$\frac{T_{z_i}}{T_{\max}}\mathrm{CSG}(\vec{z}_i) \leq \frac{\kappa_m}{\eta}\big(\ell(\vec{w}_0) - \ell(\vec{w}_*)\big) + \frac{\kappa_m\eta_s\rho\Gamma_v^2 T_{z_i}}{2}.$$

(37)

Taking expectation over $z_i$ and training randomness $p$:

$$\mathbb{E}_{z_i,p}\left[\frac{T_{z_i}}{T_{max}}\mathrm{CSG}(\vec{z}_i)\right] \leq \frac{\kappa_m}{\eta}\mathbb{E}_{t,z_i,p}\left[\ell(\vec{w}_0, \vec{z}_i) - \ell^{\setminus z_i}(\vec{w}_*, \vec{z}_i)\right]$$

$$+ \frac{\kappa_m}{\eta}\mathbb{E}_{t,z_i,p}\left[\ell^{\setminus z_i}(\vec{w}_*, \vec{z}_i) - \ell(\vec{w}_*, \vec{z}_i)\right] + \frac{\kappa_m\eta_s\rho\Gamma_v^2\,\mathbb{E}_{z_i,p}[T_{z_i}]}{2}$$

(38)

Next using the memorization result from Ravikumar et al. (2024a) (Eq. 32):

$$\mathbb{E}_{z_i,p}\left[\frac{T_{z_i}}{T_{max}}\mathrm{CSG}(\vec{z}_i)\right] \leq \frac{\kappa_m}{\eta}\mathbb{E}_{t,z_i,p}\left[\ell(\vec{w}_0,\vec{z}_i) - \ell^{\backslash z_i}(\vec{w}_*,\vec{z}_i)\right]$$

$$+ \frac{\kappa_m}{\eta}\mathbb{E}_{t,z_i}\left[L\,\mathrm{mem}(\vec{z}_i)\right] + \frac{\kappa_m\eta_s\rho\Gamma_v^2\,\mathbb{E}_{z_i,p}\left[T_{z_i}\right]}{2} \tag{39}$$

$$\mathbb{E}_{z_i,p}\left[\frac{T_{z_i}}{T_{\max}}\mathrm{CSG}(\vec{z}_i)\right] \leq \frac{\kappa_m L}{\eta}\mathbb{E}_{z_i}\left[\mathrm{mem}(\vec{z}_i)+1\right] + \frac{\kappa_m\eta_s\rho\Gamma_v^2}{2}\mathbb{E}_{z_i,p}[T_{z_i}]. \tag{40}$$

We need to separate the two terms within the expectation, for which we use the following result

$$\mathbb{E}[a(z)b(z)] = \mathbb{E}[a(z)]\,\mathbb{E}[b(z)] + \mathrm{Cov}(a(z),b(z)) \tag{41}$$

By choosing $\tau > \dfrac{\kappa_m^2\eta_s\rho\Gamma_v^2}{2\eta} \implies C_1 > 0$, and by Theorem 4.2 we know $\mathrm{Cov}(\mathrm{CSG}(\vec{z}_i),T_{z_i}) \geq 0$. Thus, it follows that

$$\mathbb{E}_{z_i,p}\left[\frac{T_{z_i}}{T_{\max}}\right]\mathbb{E}_{z_i,p}\left[\mathrm{CSG}(\vec{z}_i)\right] \leq \frac{\kappa_m L}{\eta}\mathbb{E}_{z_i}\left[\mathrm{mem}(\vec{z}_i)+1\right] + \frac{\kappa_m\eta_s\rho\Gamma_v^2}{2}\mathbb{E}_{z_i}[T_{z_i}]. \tag{42}$$

Thus,

$$\mathbb{E}_{z_i}[T_{z_i}] \leq \frac{\dfrac{\kappa_m L}{\eta}\left(\mathbb{E}_{z_i}[\mathrm{mem}(\vec{z}_i)]+1\right)}{\dfrac{\mathbb{E}_{p,z_i}[\mathrm{CSG}(\vec{z}_i)]}{T_{\max}} - \dfrac{\eta_s\kappa_m\rho\Gamma_v^2}{2}}. \tag{43}$$

For the division performed in Eq. 43 we require the denominator to be strictly positive

$$\frac{\mathbb{E}_{p,z_i}[\mathrm{CSG}(\vec{z}_i)]}{T_{\max}} - \frac{\eta_s\kappa_m\rho\Gamma_v^2}{2} \geq 0, \tag{44}$$

which implies

$$\eta_s \leq \frac{2\,\mathbb{E}_{p,z_i}[\mathrm{CSG}(\vec{z}_i)]}{T_{\max}\kappa_m\rho\Gamma_v^2} \tag{45}$$

The sign of the denominator is determined by the learning rate, $\eta_t$ (since $\eta_s$ is max learning rate over training). A sufficiently small positive learning rate can be chosen to ensure the denominator is positive.

Substituting into Equation 33 in Eq. 47 yields

$$C_a\mathbb{E}_{z_i}[T_{z_i}] \geq L\,\mathbb{E}_{z_i}[\mathrm{mem}(\vec{z}_i)] + \frac{\eta}{\kappa_m}\mathbb{E}_{p,z_i}[\mathrm{CSG}(\vec{z}_i)] - C_b. \tag{46}$$

Equivalently,

$$C_a\frac{\dfrac{\kappa_m L}{\eta}\left(\mathbb{E}_{z_i}[\mathrm{mem}(\vec{z}_i)]+1\right)}{\dfrac{\mathbb{E}_{p,z_i}[\mathrm{CSG}(\vec{z}_i)]}{T_{\max}} - \dfrac{\eta_s\kappa_m\rho\Gamma_v^2}{2}} \geq \frac{\eta}{\kappa_m}\mathbb{E}_{p,z_i}[\mathrm{CSG}(\vec{z}_i)] - C_c, \tag{47}$$

where

$$C_a = \frac{\eta\tau}{\kappa_m} - \frac{\kappa_m\eta_s\rho\Gamma_v^2}{2}, \quad C_b = \frac{\kappa_m\eta_s T_{\max}\rho\Gamma_v^2}{2} + \beta \quad C_c = C_b - L\,\mathbb{E}_{z_i}[\mathrm{mem}(\vec{z}_i)]$$

For the above division (Eq.47) to hold, we need $C_a > 0$ and denominator $> 0$, we have already shown a sufficiently small positive learning rate can be chosen to ensure the denominator is positive.

Next about $C_a$, since the learning threshold $\tau$ is arbitrary we choose $\tau$ so that, $C_a > 0$. This is true if $\tau > \dfrac{\kappa_m^2\eta_s\rho\Gamma_v^2}{2\eta}$.

Rearranging Eq. 47 as a quadratic we have

$$\frac{\eta \, \eta_s}{\kappa_m T_{max}} \mathbb{E}_{p,z_i}^2 \left[\mathrm{CSG}(\vec{z}_i)\right] - \left[\frac{C_c \eta}{T_{max}} + \frac{\eta^2 \eta_s \rho \Gamma_v^2}{2}\right] \mathbb{E}_{p,z_i} \left[\mathrm{CSG}(\vec{z}_i)\right] + \frac{\eta \, \eta_s \kappa_m \rho \Gamma_v^2 C_c}{2} \tag{48}$$

$$\leq C_a \kappa_m L \left(\mathbb{E}_{z_i}[\mathrm{mem}(\vec{z}_i)] + 1\right) \tag{49}$$

Note $\eta$ is a function of $T_{max}$. Normally for dominant term analysis we can assume $\eta \propto \sqrt{T_{max}}$. Let

$$X = \frac{\mathbb{E}_{p,z_i}[\mathrm{CSG}(\vec{z}_i)]}{T_{\max}}. \tag{50}$$

Then the inequality takes quadratic form

$$aX^2 + bX + c \leq 0, \tag{51}$$

with

$$a = \frac{\eta \eta_s}{\kappa_m}, \quad b = -\left(\frac{C_c \eta}{T_{\max}^2} + \frac{\eta^2 \eta_s \rho \Gamma_v^2}{2 T_{\max}}\right), \quad c = \frac{\eta \eta_s \kappa_m \rho \Gamma_v^2 C_c}{2 T_{\max}} - \frac{C_a \kappa_m L}{T_{\max}}\left(\mathbb{E}_{z_i}[\mathrm{mem}(\vec{z}_i)] + 1\right). \tag{52}$$

By the quadratic formula, the inequality holds if

$$r_1 \leq X \leq r_2, \tag{53}$$

where $r_1, r_2$ are the roots.

Since $C_a \propto \eta$, $C_c \propto \mathbb{E}_{z_i}[\mathrm{mem}(\vec{z}_i)]$, and typically $\eta \propto \sqrt{T_{\max}}$, we have

$$\frac{b}{2a} = O\left(\frac{\mathbb{E}_{z_i}[\mathrm{mem}(\vec{z}_i)]}{\sqrt{T_{\max}}}\right), \qquad \frac{\sqrt{4ac}}{2a} = O\left(\frac{\mathbb{E}_{z_i}[\mathrm{mem}(\vec{z}_i)]}{\sqrt{T_{\max}}}\right). \tag{54}$$

Thus the dominant contribution is linear, yielding

$$\mathbb{E}_{p,z_i}[\mathrm{CSG}(\vec{z}_i)] = O\left(\mathbb{E}_{z_i}[\mathrm{mem}(\vec{z}_i)]\sqrt{T_{\max}}\right). \tag{55}$$

Often $T_{max}$ is fixed thus we can write the results as

$$\boxed{\mathbb{E}_{p,z_i}[\mathrm{CSG}(\vec{z}_i)] = O\left(\mathbb{E}_{z_i}[\mathrm{mem}(\vec{z}_i)]\right).} \tag{56}$$

**Conditon for real roots.** Note from Ghadimi & Lan (2013) we have $\sum_{t=0}^{T-1}\left(\eta_t - \frac{\rho \eta_t^2}{2}\right) = \eta \geq 0$.

For $b^2 - 4ac \geq 0$ we need $c \leq 0$. This is true if $C_c \leq 0$ or if $C_c \geq 0$ then, we know from the quadratic

$$c = \frac{\eta \, \eta_s \kappa_m \rho \Gamma_v^2 C_c}{2 T_{max}} - \frac{C_a \kappa_m L}{T_{max}}\left(\mathbb{E}_{z_i}[\mathrm{mem}(\vec{z}_i)] + 1\right) \tag{57}$$

$$c \leq 0 \tag{58}$$

$$\frac{\eta \, \eta_s \kappa_m \rho \Gamma_v^2 C_c}{2 T_{max}} - \left[\frac{\eta \tau}{\kappa_m} - \frac{\kappa_m \eta_s \rho \Gamma_v^2}{2}\right] \frac{\kappa_m L}{T_{max}}\left(\mathbb{E}_{z_i}[\mathrm{mem}(\vec{z}_i)] + 1\right) \tag{59}$$

$$\eta_s \left[\frac{\kappa_m \eta \rho \Gamma_v^2 C_c}{2 T_{max}} + \frac{\kappa_m^2 \rho \Gamma_v^2 L(\mathbb{E}_{z_i}[\mathrm{mem}(\vec{z}_i)] + 1)}{2 T_{max}}\right] \leq \frac{\eta \tau L(\mathbb{E}_{z_i}[\mathrm{mem}(\vec{z}_i)] + 1)}{T_{max}} \tag{60}$$

$$\eta_s \left[\kappa_m \eta \rho \Gamma_v^2 C_c + \kappa_m^2 \rho \Gamma_v^2 L(\mathbb{E}_{z_i}[\mathrm{mem}(\vec{z}_i)] + 1)\right] \leq 2\eta \tau L(\mathbb{E}_{z_i}[\mathrm{mem}(\vec{z}_i)] + 1) \tag{61}$$

Thus we have

$$\eta_s \left[\kappa_m \eta \rho \Gamma_v^2 C_c + \kappa_m^2 \rho \Gamma_v^2 L(\mathbb{E}_{z_i}[\mathrm{mem}(\vec{z}_i)] + 1)\right] \leq 2\eta \tau L(\mathbb{E}_{z_i}[\mathrm{mem}(\vec{z}_i)] + 1) \tag{62}$$

$$\eta_s \leq \frac{2\eta \tau L(\mathbb{E}_{z_i}[\mathrm{mem}(\vec{z}_i)] + 1)}{\left[\kappa_m \eta \rho \Gamma_v^2 C_c + \kappa_m^2 \rho \Gamma_v^2 L(\mathbb{E}_{z_i}[\mathrm{mem}(\vec{z}_i)] + 1)\right]} \tag{63}$$

$$\eta_s \leq \frac{2\tau L(\mathbb{E}_{z_i}[\mathrm{mem}(\vec{z}_i)] + 1)}{\kappa_m \rho \Gamma_v^2 \left[C_c + \frac{\kappa_m L}{\eta}(\mathbb{E}_{z_i}[\mathrm{mem}(\vec{z}_i)] + 1)\right]} \tag{64}$$

**Final Learning Rate condition.** For a sufficiently small positive learning rate, the condition $b^2 - 4ac > 0$ will hold true. Therefore, a suitable choice of $\eta_t$ exists, given by combination of results from Eq. 45 and Eq 64

$$0 \leq \eta_t \leq \sqrt{\eta_s} \leq \min\left(\sqrt{\frac{2\tau L(\mathbb{E}_{z_i}[\text{mem}(\vec{z_i})]+1)}{\kappa_m \rho \Gamma_v^2 \left[C_c + \frac{\kappa_m L}{\eta}(\mathbb{E}_{z_i}[\text{mem}(\vec{z_i})]+1)\right]}}, \sqrt{2\frac{\mathbb{E}_{p,z_i}[\text{CSG}(\vec{z_i})]}{T_{max}\kappa_m \rho \Gamma_v^2}}\right) \quad (65)$$

**Conclusion.** This completes the proof. That is,

$$\mathbb{E}_{p,z_i}[\text{CSG}(\vec{z_i})] = O\big(\mathbb{E}_{z_i}[\text{mem}(\vec{z_i})]\big), \quad (66)$$

subject to the step-size constraints above.

## D    LICENSES FOR ASSETS USED

For each of the assets used we present the licenses below we also have provided the correct citation in the main paper as well as here for convenience.

1. ImageNet (Russakovsky et al., 2015): Terms of access available at `https://image-net.org/download.php`
2. CIFAR10 (Krizhevsky et al., 2009): Unknown / No license provided
3. CIFAR100 (Krizhevsky et al., 2009): Unknown / No license provided
4. Pre-trained ImageNet models and code: We used pre-trained ImageNet models from (Feldman & Zhang, 2020) which is licensed under Apache 2.0 `https://github.com/google-research/heldout-influence-estimation/blob/master/LICENSE`.
5. Baseline methods: We re-implemented the baseline methods hence is provided with along with our code which is distributed under the MIT License.
6. Pytorch (Ansel et al., 2024): Custom BSD-style license available at `https://github.com/pytorch/pytorch/blob/main/LICENSE`.
7. ResNet Model Architecture (He et al., 2016): MIT license available at `https://github.com/kuangliu/pytorch-cifar/blob/master/LICENSE`

## E    COMPUTE RESOURCES

All of the experiments were performed on a heterogeneous compute cluster consisting of 9 1080Ti's, 6 2080Ti's and 4 A40 NVIDIA GPUs, with a total of 100 CPU cores and a combined 1.2 TB of main system memory. However, the results can be replicated with a single GPU with as minimal as 8GB of VRAM.

## F    LLM USAGE

The use of LLMs was limited to grammar and spelling fixes. LLMs were not used for any other part of the research project.

## G    LIMITATIONS

Compared to other memorization proxies, the limitations of CSG are minimal. Its primary constraint is access to the training process. This is not unique, and is shared by other proxies such as curvature Garg & Roy (2023), loss sensitivity Arpit et al. (2017), and forgetting events Toneva et al. (2019). This requirement is also common among methods for identifying mislabeled examples Maini et al. (2022); Northcutt et al. (2021).

## H    BROADER IMPACT

This work introduces CSG, an extremely efficient proxy for per-example memorization in deep networks that matches stability-based scores while being up to 5 orders of magnitude cheaper to compute. Because CSG can be gathered during training and aligns with memorization, learning time, and input curvature, it becomes a practical lens through which practitioners and, potentially, regulators can watch how modern models absorb, remember, and sometimes over-fit their data.

- **Better generalization and safer deployment.** By flagging the exact examples that a model memorizes, CSG helps teams diagnose over-fitting before deployment and choose an early-stopping point without a validation set, cutting training epochs.

- **Higher-quality, fairer datasets.** The method surfaces mislabeled, duplicate, and biased samples achieving state-of-the-art detection accuracy on CIFAR-10/100 and Fashion-MNIST. Removing or relabeling such data can reduce spurious correlations, improving fairness for under-represented sub-populations.

- **Privacy auditing and mitigation.** CSG strongly correlates with membership-inference success and adversarial distance. Early-stopping at the CSG peak reduces a suite of membership-inference attacks relative to the final checkpoint, making CSG both a *privacy-risk indicator* and a knob for mitigating leakage without extra noise injection.

