# OpenReview forum: "Memorization Through the Lens of Sample Gradients"
_ICLR.cc/2026/Conference — ICLR 2026 Poster_

### Official Review · Reviewer_maBN · 2025-10-30

**Soundness:** 4
**Presentation:** 3
**Contribution:** 3
**Rating:** 6
**Confidence:** 4

**Summary:**

This work proposes a computationally efficient way to approximate the degree of memorization in deep neural nets. Based on the observation that memorized samples tend to have longer training time, this work proposes cumulative sample gradient (CSG) as a proxy for memorization. Theoretical results show the relation between CSG and learning time & memorization. Empirical evaluations corroborates the findings and shows superior computational performance over previous state-of-the-art.

**Strengths:**

Overall, this work is well-written and organized. It motivates the problem settings and draws fairly clear connection with previous work. The contribution is pertinent to the current challenge of ML. By providing a more efficient probe for the phenomenon of memorization, this work can accelerate future research in this field. The theoretical formulations are solid and with clear purpose: they do shed light into the construction of the practical proxy. The empirical improvements are encouraging. No major flaw with experiment design.

**Weaknesses:**

The work doesn't have apparent weaknesses that might lead to clear reject. I do have a few questions about the assumption and the source of computation edge over previous work. Having them clarified can better help the reader understand the contribution and use the tool with confidence.

There are a few grammatic glitches here and there. For example, "is plays" at Line 310 and "it's roots" at Line 258. Can be fixed by proof reading.

**Questions:**

1) What makes the computation of CSG faster than CSL? Seems that both metrics are cumulative and the computation of loss/gradient are not too different in general. Could you tell us more about the source of speedup?

2) The theoretical results show that CSG is upper bounded by learning time and memorization. Does that mean high CSG -> high memorization? What about the opposite direction, does low CSG -> low memorization?

3)  The theories are formulated against pure SGD. Does CSG reliably detect memorization/mislabeled samples for different optimizers? If time allows, could you show some evidence of CSG's success for other optimizer?

4) Are previous SOTA's performance dependent on the choice of optimizer?

---

> ### Author Response · Authors · 2025-11-21
>
> We thank the reviewer for their feedback and questions. We have addressed the concerns raised regarding baselines, architectural scope, and experimental details below. Additionally, we have provided new baselines and experimental results (including ViT experiments and empirical constant estimations) at the following anonymous link with in compliance with ICLR guidelines: **[https://lively-mushroom-060927110.3.azurestaticapps.net/](https://lively-mushroom-060927110.3.azurestaticapps.net/)**
>
> Q. There are a few grammatic glitches here and there. For example, "is plays" at Line 310 and "it's roots" at Line 258. Can be fixed by proof reading.
>
> **A:** Thank you for bringing this to our notice, we will fix this in the revised version of the paper.
>
> **Questions:**
>
> 1. What makes the computation of CSG faster than CSL? Seems that both metrics are cumulative and the computation of loss/gradient are not too different in general. Could you tell us more about the source of speedup?
>
> 	**A:** Our major finding is that the peak input gradient which correlates with memorization occurs early in training, which is only possible to track via average input gradient. This allows us to *stop training early* (between 10% - 30% of training) to calculate the memorization proxy CSG and SGAL, instead of competing methods needed to train to completion.
>
> 2. The theoretical results show that CSG is upper bounded by learning time and memorization. Does that mean high CSG -> high memorization? What about the opposite direction, does low CSG -> low memorization?
>
> 	**A:** While the theoretical results do not explicitly derive the "CSG -> memorization" link, our empirical results show that the bound is likely tight. Together with this empirical evidence, the results suggest a strong link between CSG and memorization, indicating that CSG is a useful proxy for memorization.
>
> 3. The theories are formulated against pure SGD. Does CSG reliably detect memorization/mislabeled samples for different optimizers? If time allows, could you show some evidence of CSG's success for other optimizer?
>
> 	**A:** This was a common question raised by reviewers. We have collected results on Adagrad, RMSProp, Adam, and AdamW, and we observe the same trends as with SGD. We provide an interactive website to visualize the training behavior and input gradient validation loss across various optimizers here: https://lively-mushroom-060927110.3.azurestaticapps.net/?tab=mem-sim
>
> 4. Are previous SOTA's performance dependent on the choice of optimizer?
>
> 	**A:** Results on various optimizers are also provided, which clearly show that the performance of prior SOTA works is not dependent on the choice of optimizer. Please see the table on our rebuttal website: [https://lively-mushroom-060927110.3.azurestaticapps.net/?tab=mem-sim](https://lively-mushroom-060927110.3.azurestaticapps.net/?tab=mem-sim)

---

### Official Review · Reviewer_BKpC · 2025-10-30

**Soundness:** 2
**Presentation:** 2
**Contribution:** 2
**Rating:** 2
**Confidence:** 5

**Summary:**

The paper proposes CSG, a fast, theoretically grounded proxy for measuring memorization in deep networks by accumulating input loss gradients during training. CSG correlates strongly with true memorization scores while being up to five orders of magnitude more efficient, enabling validation-free early stopping, mislabeled-sample detection, and bias discovery.

**Strengths:**

CSG offers a theoretically grounded and computationally efficient way to estimate memorization, achieving near-perfect correlation with true scores at a fraction of the cost.

It enables validation-free early stopping and state-of-the-art mislabeled data detection, making it both practical and interpretable for large-scale deep learning.

**Weaknesses:**

Novelty:
My primary concern lies in the novelty of the work. The authors’ main observation that memorization tends to occur in the later stages of training is well established and has been extensively documented in prior studies [1,2]. Likewise, leveraging gradients to approximate or track memorization has been explored before [3,4], making the core idea appear incremental rather than groundbreaking. Can the authors show how their work performs in relation to [1,3,4].

Computational Cost:
While the proposed approach claims efficiency, computing cumulative sample gradients still requires forward and backward passes for each sample at every epoch, which can be prohibitive for large models and datasets. Prior work [3] already proposes strategies to reduce this overhead.

Limited Optimizer Evaluation:
The experiments rely solely on the Adam optimizer. To demonstrate broader applicability, results should be validated across multiple optimizers such as SGD, RMSProp, and AdamW. Can authors provide more info on how their method would behave across optimizers?


[1]. Agiollo, Andrea, Young In Kim, and Rajiv Khanna. "Approximating Memorization Using Loss Surface Geometry for Dataset Pruning and Summarization." Proceedings of the 30th ACM SIGKDD Conference on Knowledge Discovery and Data Mining. 2024.
[2] https://aclanthology.org/2024.blackboxnlp-1.4
[3] https://arxiv.org/abs/2008.11600
[4] https://arxiv.org/pdf/2002.08484

**Questions:**

Above

---

> ### Author Response · Authors · 2025-11-21
>
> ### Part (1/2)
> We thank the reviewer for their feedback. We appreciate the opportunity to clarify the distinct contributions of our method compared to prior literature and to highlight the significant efficiency gains our approach offers. Additionally, we have provided new baselines and experimental results at the following anonymous link with in compliance with ICLR guidelines: **[link](https://lively-mushroom-060927110.3.azurestaticapps.net/)**
>
> ### 1. Novelty and Comparison to Prior Work
>
> While we acknowledge that gradient-based analysis has been explored in the literature, our work fundamentally differs from the cited studies in terms of scope, objective, and empirical efficacy
>
> Comparison with Agiollo et al. (SAMIS) [1]: The primary distinction lies in static vs dynamics and computational cost.
> - Dynamics vs. Static: [1] analyzes the performance gap between SGD and SAM trained models on the validation set at the end of training. Our work focuses on training dynamics.
> - Furthermore, we respectfully express reservations regarding the premise in [1] that the discrepancy between SAM and SGD acts as a proxy for intrinsic "data atypicality." Memorization is inherently algorithm-dependent; it is a function of both the dataset and the algorithm. Because SAM introduces a distinct inductive bias compared to SGD, it alters the specific subset of examples that are memorized. Consequently, using the performance gap between two distinct algorithms to define memorization results in low correlation between SAMIS and FZ score shown in **[Table Anon Link](https://lively-mushroom-060927110.3.azurestaticapps.net/?tab=mem-sim)**. Our work avoids this confounding factor by analyzing the trajectory of a single training run.
> - Computational Cost: [1] rely on a k-fold cross-validation framework (they heuristically choose 9-fold). Their method requires training with both SGD and SAM. This results in a computational cost approximately 24x higher (~8x from folds since each fold is 0.9x and 3x for each fold since SGD is 1x and SAM ~2x) than a standard training run. In contrast, our method requires 0.3x of a single standard training run. Thus our method is 80x faster than SAMIS.
>
> Comparison with [2]: This work focuses exclusively on language models and _does not track input gradients_. Their analysis relies on exact sequence matches in the training data. While they draw similar conclusions about memorization dynamics, our gradient-based insight (the rise-peak-decline pattern) is novel to our work and is robust across various optimizers and weight decay settings (see results [here](https://lively-mushroom-060927110.3.azurestaticapps.net/?tab=wd&opt=adagrad&data=ig%2Cvl)), enabling broad applicability. We also provide deeper theoretical and quantitative analysis compared to [2] ([2] do not provide any theory), including a comparison of our metric with Feldman-Zhang's version along with various baselines and applications where such proxies are useful (mislabel detection and bias discovery).
>
> Comparison with Variance of Gradients (VoG) [3]: While VoG also utilizes input gradients, the underlying mechanism and utility differ significantly:
>
> - Metric Difference: VoG tracks the _variance_ of gradients to identify data quality issues (e.g., mislabeled samples). Our method (CSG/SGAL) tracks cumulative gradients.
> - Performance: As shown in **[Table Anon Link](https://lively-mushroom-060927110.3.azurestaticapps.net/?tab=mem-sim)** , VoG is a suboptimal proxy for memorization. We compared both "All-Checkpoint VoG" and "Last-5 Checkpoint VoG" against the FZ memorization score. We attribute the lower performance to the fact that for VoG [3], a sample that is consistently hard (high memorization) has low variance (yielding a low VoG score) but high accumulation (yielding a high CSG score). This justifies the performance gap between CSG and VoG as memorization proxies. Our results, shown in **[Table Anon Link](https://lively-mushroom-060927110.3.azurestaticapps.net/?tab=mem-sim)**, demonstrate a significantly higher correlation with the FZ score than VoG.
>
> Comparison with TracIn [4]: TracIn is rooted in estimating influence functions. While self-influence is theoretically related to memorization, it is not a drop-in
>
> - TracIn uses gradients in the parameter space, which needs single sample gradient estimate which are extremely slow compared to our input gradient which can be estimated using standard mini-batch backward pass with almost no overhead.
> - We provide a self-influence baseline using TracIn in our experiments. Our results demonstrate that while TracIn captures some memorization signals, it does not perform as well as our input gradient tracking method. Our approach offers a more direct and empirically stronger signal for identifying memorized samples.
>
> We have added all these baseline in  **[Table Anon Link](https://lively-mushroom-060927110.3.azurestaticapps.net/?tab=mem-sim)** and provided results across various optimizers.

---

> ### Author Response · Authors · 2025-11-21
>
> ### Part (2/2)
>
> ### 2. Computational Cost & Efficiency
>
> In fact, our approach offers upto **3x efficiency gain** over methods like VoG [3]
>
> - **The Early Stopping Advantage:** The reviewer correctly notes that [3] saves costs by using only the _last_ few checkpoints. However, this still requires training the model to **completion** (100% of epochs) before the analysis can begin (so no training cost savings at all since we need to train to 100%). The authors recommend using the last 3 checkpoint (See Section 2.2, Number of checkpoints in [3]). In contrast our major finding is that the peak input gradient which correlates with memorization occurs early in training, which is only possible to track via average input gradient. This allows us to **stop training early** (between 10% - 30% of training) to calculate the memorization score proxy CSG and SGAL.
>
> - **Overhead is Negligible:** On modern hardware training on CIFAR100/ImageNet, the overhead of calculating input gradients is measurably negligible when training the model, we measured both with input grad calculation and vanilla training both to take 23 seconds per epoch on our hardware.
>
> Additionally, reference [1] is computationally demanding, costing approximately 24x more than vanilla training and 80x more than our proposed SGAL and CSG methods. Similarly, TracIn [4] relies on single-sample weight gradients, which are experimentally expensive. In our benchmarks, TracIn required 10 minutes per epoch, whereas our method took only 23 seconds a 26x reduction in per-epoch computational cost. These runs were measured on the same machine with identical batxh size and settings. Furthermore, because CSG allows for early stopping (at 30% of training), the total efficiency gain increases to 87x compared to TracIn. This performance advantage arises because our method avoids the inefficient batch-size-of-one operations required to estimate single-sample loss and gradients, instead utilizing mini-batch gradients to estimate single-sample input gradients. We will update the revised version of the paper to include these additional baselines and compute costs.
>
> ### 3. Limited Optimizer Evaluation:
>
> We appreciate this feedback. To clarify our results are on SGD in the paper, but to address this concern (which was shared by other reviewers), we have added results using RMSProp, AdamW, Adagrad, and Adam. You can examine the performance of these optimizers via our **[interactive visualization](https://lively-mushroom-060927110.3.azurestaticapps.net/?tab=wd&opt=adagrad&data=ig%2Cvl)**. We also provide results on cosine scheduled (cosine annealing lr update) learning on ImageNet with ViT and ResNet50 **[plot link](https://lively-mushroom-060927110.3.azurestaticapps.net/?tab=imgnet-csg)** and see the same trend with the rise peak-decline of input gradient and this was trained using pytorch ImageNet model training (timm library), to maximize reproducibility on state of the art training recipes.

---

### Official Review · Reviewer_gxif · 2025-10-30

**Soundness:** 3
**Presentation:** 3
**Contribution:** 3
**Rating:** 6
**Confidence:** 3

**Summary:**

The paper proposes Cumulative Sample Gradient (CSG)—the loss gradient w.r.t. the input, accumulated over training—as a computationally cheap proxy for stability-based memorization (Feldman & Zhang, 2020). The authors provide theory that (i) expected CSG is upper-bounded by learning time (Theorem 4.2) and (ii) linearly bounded by memorization (Theorem 4.3). Empirically, they observe a characteristic rise–peak–decline trajectory for average per-sample input gradients that aligns with a peak in weight norm and the first minimum in validation loss (double-descent boundary), enabling validation-free early stopping. They also introduce SGAL (a loss accumulated only until the gradient-based stopping point) for further efficiency. Across CIFAR-100/ImageNet, CSG/SGAL correlates well with memorization scores, is substantially faster than curvature and CSL proxies, supports mislabeled-sample detection, and helps surface dataset biases.

**Strengths:**

•	Originality & clarity: Using input gradients accumulated over training as a memorization proxy is elegant; the rise–peak–decline alignment with the weight norm and double-descent boundary is compelling and clearly presented.
•	Quality (theory): Theorems relating CSG to learning time and memorization provide formal grounding absent in many proxies; assumptions and proof sketches are transparent.
•	Quality (empirics): Consistent binned linear trends; strong correlation with F&Z scores; broad comparisons (CSL, curvature, forgetting events, loss sensitivity) on CIFAR-100/ImageNet; MIA and adversarial-distance analyses support the privacy link.
•	Significance & practicality: Large speedups (0.1–0.3× of standard training vs. 3.6–14.3× for curvature; orders of magnitude vs. F&Z) lower the barrier to dataset diagnostics at scale; mislabeled-sample AUROCs are SOTA or competitive at all noise levels.

**Weaknesses:**

•	Assumption sensitivity and constant opacity. The theoretical bounds rely on β-stability, Lipschitz continuity, L-bounded losses, and learning-rate conditions, and exclude first-layer skip connections. Constants involving the pseudo-inverse of batch matrices (κ terms) may be large/ill-conditioned, making the bounds hard to interpret quantitatively.
•	Calibration claim is mixed. Table 2 shows lower ECE for the last epoch (0.1017) than for gradient-based stopping (0.1382), contradicting the blanket statement that early-stopped checkpoints have lower calibration errors; other metrics (MCE/MSCE/UCE) favor earlier stopping, so the narrative should be nuanced.
•	Scope of datasets/models. Results are primarily on CIFAR-100/ImageNet with ResNet/Inception. It would strengthen generality to include modern architectures (e.g., ViT) and tasks beyond image classification, since input-gradient behavior and training dynamics may differ. (The Adam experiment is a useful first step.)
•	Comparative coverage. While CSL/curvature/forgetting/loss-sensitivity are included, some adjacent proxies (e.g., EL2N, GraNd / importance-sampling-style difficulty measures) and influence-based approximations (e.g., TracIn) are not compared; these could provide a more complete picture of trade-offs.
•	Training-access requirement. Like many proxies, CSG needs access to per-sample gradients during training; this limits pure post-hoc auditing scenarios (the limitation is acknowledged).
•	Qualitative bias analysis. The bias discovery examples are informative but largely qualitative; quantitative fairness metrics (e.g., subgroup error rates) would make the case stronger.

**Questions:**

1.	How sensitive is the weight-norm peak rule to weight decay, label smoothing, data augmentation strength, and optimizer hyperparameters? A small ablation across these knobs would clarify robustness.
2.	Practically, do you compute input gradients every iteration for all samples, or on a schedule/subset? Please quantify wall-clock overhead vs. vanilla training across model sizes.
3.	Have you tested CSG/SGAL on ViTs or transformers for NLP? If not, what obstacles (e.g., tokenization, augmentation) do you anticipate?
4.	Can you empirically estimate the constants in Lemma 4.1 (e.g., behavior of κ through training) to illustrate why the linear trends emerge despite potential ill-conditioning?
5.	Given Table 2 shows mixed results across ECE/MCE/MSCE/UCE, can you reconcile the claim “lower calibration errors than last epoch” and specify which metrics you prioritize and why?
6.	Since you use precomputed F&Z scores, how sensitive are your correlations to training recipe variations (architectures different from F&Z)?

---

> ### Author Response · Authors · 2025-11-21
>
> ### Part (1/2)
>
> We thank the reviewer for their feedback and insightful questions. We have addressed the concerns raised regarding baselines, architectural scope, and experimental details below. Additionally, we have provided new baselines and experimental results (including ViT experiments and empirical constant estimations) at the following anonymous link with in compliance with ICLR guidelines: **[https://lively-mushroom-060927110.3.azurestaticapps.net/](https://lively-mushroom-060927110.3.azurestaticapps.net/)**
>
> ---
> ### **Response to Weaknesses**
>
> **1. Calibration claim is mixed (ECE vs. MCE/UCE).**
>
> **Response:** This is an excellent observation. Indeed, ECE follows the trend noted by the reviewer. However, metrics like MCE and UCE favor our early-stopping method. We prioritize MCE and UCE in this context because real-world data distributions often follow a **long-tail / power-law distribution**. As discussed by Feldman and Zhang [1], deep learning models rely on memorizing these rare, atypical examples (the "long tail") to achieve generalization. ECE is an expectation-based metric; it weights the calibration error of each bin by the number of samples in that bin. Consequently, in long-tailed datasets, low-accuracy bins often contain fewer samples (the "tail") and are down-weighted by ECE, masking miscalibration on difficult examples. In contrast, MCE and UCE treat the calibration across the accuracy range more equally. When the goal is trustworthy AI that performs reliably on both head and tail data, these metrics provide a more accurate reflection of model safety.
>
> **2. Scope of datasets/models (Need for ViT/Modern architectures).**
>
> **Response:** We agree that testing on modern architectures strengthens the generality of our claims. We have extended our evaluation to include **Vision Transformers (ViT)** on ImageNet. We observed results consistent with our findings for CNNs (ResNet/Inception). These additional results can be found at the **[anonymous link](https://lively-mushroom-060927110.3.azurestaticapps.net/?tab=imgnet-csg)**.
>
> **3. Comparative coverage (Missing GraNd, TracIn, etc.).**
>
> **Response:** We have updated our analysis to include comparisons with **GraNd** and **TracIn**. We have added these methods to our memorization similarity score evaluations to provide a more complete picture of the trade-offs.
>
> **4. Training-access requirement (Per-sample gradients).**
>
> **Response:** We would like to clarify that our method **does not** require the computation of expensive per-sample gradients in the same manner as influence functions like TracIn or GraNd. Our method utilizes input gradients that can be computed efficiently during standard mini-batch training (see our response to **Question 2** for runtime details).
>
> **6. Assumption sensitivity.**
>
> **Response:** Please refer to our response to **Question 4**, where we empirically estimate the condition number $\kappa$ throughout training to demonstrate stability.
>
> ### **Response to Questions**
>
> **1. How sensitive is the weight-norm peak rule to hyperparameters (weight decay, label smoothing, etc.)?**
>
> R: We performed the requested ablation study across various optimizers and weight decay values. We found that the peak weight norm is significantly less robust to hyperparameter changes compared to the peak input gradient.
>
> The peak input gradient consistently occurs at the same training stage across different settings. In contrast, the weight norm peak varies. The results provided in our anonymous link show the maximum and average $\kappa$ across iterations; the empirical results suggest that the input gradient is a far more reliable signal for stopping.
>
> **2. Practically, do you compute input gradients every iteration for all samples? Quantify overhead.**
>
> R: Input gradients are computed for all samples at every iteration for the entire training set. However, because this is integrated into standard mini-batch training, the computational cost is negligible.
>
> Wall-clock comparison: On a GTX 1080Ti with an Inception model (batch size 256):
> - **Vanilla Training:** ~22.9 seconds per epoch
> - **Training with CSG:** ~22.9 seconds per epoch
>
> Essentially even on older GTX1080Ti we obtain the signal "for free" during the backward pass.
>
> **3. Have you tested CSG/SGAL on ViTs or transformers for NLP?**
>
> **R:** We have included ViT results (see Weakness 2), but we have not yet tested CSG or SGAL on NLP tasks. We anticipate two main differences/challenges for NLP:
>
> 1. **Lack of Baselines:** Unlike vision models, where "memorization scores" (via leave-one-out retraining) are pre-computed and established by Feldman and Zhang, establishing these baselines for large language corpora requires a prohibitive compute budget.
>
> 2. **Technical Adaptation:** Modern LLMs typically see data only once (single epoch) rather than multi-epoch training, and the input gradient must be computed with respect to the **input embeddings** rather than discrete tokens.

---

> ### Author Response · Authors · 2025-11-21
>
> ### Part (2/2)
>
> **4. Can you empirically estimate the constants in Lemma 4.1 (e.g., behavior of $\kappa$)?**
>
> **Response:** We empirically estimated $\kappa$ on CIFAR-100 using various optimizers. The results, shown in the table below, demonstrate that $\kappa$ remains relatively stable.
>
> |**Optimizer**|**Max κ**|**Avg κ**|
> |---|---|---|
> |**Adagrad**|1857.15|24.83|
> |**AdamW**|538.61|4.42|
> |**Adam**|674.05|6.43|
> |**RMSprop**|1034.56|6.70|
>
> **5. Can you reconcile the claim “lower calibration errors than last epoch” regarding ECE/MCE/MSCE/UCE?**
>
> **Response:** Please refer to our response to **Weakness 1**, where we explain why MCE/UCE are better suited for measuring calibration on long-tailed distributions compared to ECE.
>
> **6. How sensitive are your correlations to training recipe variations (architectures different from F&Z)?**
>
> **Response:** We have conducted additional experiments using architectures different from those used by Feldman and Zhang. These results confirm that our correlations hold across different model structures. The detailed results are available at the **[anonymous link](https://lively-mushroom-060927110.3.azurestaticapps.net/?tab=arch-mem)**. We see the best correlation is when the architectures match.
>
> ---
>
> [1] Feldman, V., & Zhang, C. (2020). What neural networks memorize and why: Discovering the long tail via influence estimation.

---

### Official Review · Reviewer_P7vQ · 2025-10-31

**Soundness:** 2
**Presentation:** 3
**Contribution:** 3
**Rating:** 6
**Confidence:** 3

**Summary:**

The paper proposes Cumulative Sample Gradient (CSG) as a theoretically motivated and computationally efficient proxy for memorization in deep neural networks. The authors define CSG as the gradient of the loss with respect to input samples, accumulated across training, and claim it correlates strongly with Feldman & Zhang’s formal memorization score while being orders of magnitude cheaper to compute. They theoretically show that CSG is bounded by both learning time and memorization, then empirically validate this relation across CIFAR-100 and ImageNet. Moreover, they propose that the peak of the model’s weight norm corresponds to the optimal early-stopping point, eliminating the need for a validation set. The paper also introduces Sample Gradient Assisted Loss (SGAL) as an efficiency improvement, and reports strong performance on tasks such as mislabeled sample detection and dataset bias discovery

**Strengths:**

1. The work links input-space gradients to memorization and learning dynamics through formal theorems, extending prior work that primarily focused on weight gradients or loss-based proxies.

2. The idea of accumulating input gradients incurs minimal additional computation during training and is potentially useful for large-scale data auditing, noisy-label detection, and privacy diagnostics.

3. The observed “rise–peak–decline” trajectory in sample gradients and weight norms provides an intuitive link between optimization dynamics and generalization behavior.

**Weaknesses:**

1. The main claim that “Cumulative Sample Gradient” represents a gradient of loss with respect to the input, accumulated over training, is conceptually questionable. The proposed CSG is essentially an aggregated gradient norm trajectory rather than a true differentiable functional of the loss. Treating it as a gradient object conflates sensitivity analysis (∇ₓℓ) with memorization, which lacks theoretical grounding in generalization theory. The derivations (Theorems 4.2–4.3) merely establish loose proportionality bounds without proving causality or sufficiency.

2. The assertion that the peak of weight norm universally coincides with the minimum validation loss is overstated. This correspondence may depend on architecture, optimizer, and regularization strength, and may fail under strong augmentation or non-stationary data.

3. While the authors claim that CSG generalizes across tasks, they only test standard supervised image classification. The theoretical link assumes uniform β-stability of SGD and bounded loss, which rarely holds in modern deep nets. It remains unclear whether CSG maintains predictive utility in other regimes such as self-supervised, generative, or multi-label settings.

**Questions:**

1. Since CSG is defined as the accumulated input gradient norm, not the derivative of a loss functional over training trajectories, do we have a rigorous reason to treat it as a “gradient of loss with respect to input samples, accumulated over training”? How does this differ from simply tracking cumulative sensitivity?

2. The paper asserts that stopping at the peak weight norm matches the minimum validation loss. Can this be proven under general conditions? How robust is this correspondence across architectures, datasets, or optimizers (e.g., Adam, Adagrad, adaptive schedulers)?

3. Do you think the CSG–memorization relationship could generalize to regression, contrastive, or generative models, where accuracy or label definitions differ? You don’t need to perform new experiments—rather, please share your intuition on whether and why such generalization might hold.

---

> ### Author Response · Authors · 2025-11-21
>
> ### Part (1/2)
> We thank the reviewer for their feedback and insightful questions. We have addressed the concerns raised regarding baselines, architectural scope, and experimental details below. Additionally, we have provided new baselines and experimental results (including ViT experiments and empirical constant estimations) at the following anonymous link with in compliance with ICLR guidelines: **[https://lively-mushroom-060927110.3.azurestaticapps.net/](https://lively-mushroom-060927110.3.azurestaticapps.net/)**
>
> 1. The main claim that “Cumulative Sample Gradient” represents a gradient of loss with respect to the input, accumulated over training, is conceptually questionable. The proposed CSG is essentially an aggregated gradient norm trajectory rather than a true differentiable functional of the loss. Treating it as a gradient object conflates sensitivity analysis (∇ₓℓ) with memorization, which lacks theoretical grounding in generalization theory. The derivations (Theorems 4.2–4.3) merely establish loose proportionality bounds without proving causality or sufficiency.
>
>      **A:**  The connection between generalization and gradient norm is indeed interesting. First, Ravikumar et al. [7] proved that the input gradient is upper-bounded by the weight gradient, and that the input gradient converges similarly to the weight gradient (i.e., optimizing in the weight space also optimizes in the input space). Furthermore, prior works [2] link the quality of the SGD solution to generalization. Since the input gradient norm is linked to the weight gradient norm which is, in turn, linked to generalization there is a potential framework to explore the link between input sensitivity and generalization. We agree with the reviewer that the theoretical results are bounds; however, we validate these bounds in practice by providing two useful applications: a proxy for memorization (identifying mislabeled samples) and a qualitative analysis of the ability to identify biases.
>
> 2. The assertion that the peak of weight norm universally coincides with the minimum validation loss is overstated. This correspondence may depend on architecture, optimizer, and regularization strength, and may fail under strong augmentation or non-stationary data.
>
>      **A:**  Prompted by similar comments from other reviewers, we performed an analysis of this behavior across various optimizers and architectures. Please refer to this **[interactive plot](https://lively-mushroom-060927110.3.azurestaticapps.net/?tab=wd&opt=adagrad&data=ig%2Cvl)**, which illustrates results across different weight decay settings and optimizers. Additionally, **[this link](https://lively-mushroom-060927110.3.azurestaticapps.net/?tab=imgnet-csg)** presents results for ViT and ResNet50 on ImageNet using a cosine scheduler (via the timm library with augmentation for SoTA training). We found that the input gradient is a more robust indicator than the weight norm for identifying the early stopping point corresponding to the lowest validation loss. As noted in the manuscript (section 4.3 Practical takeaway) , due to the sensitivity of the weight norm, we suggest monitoring both the input gradient and the weight norm.
>
> 3. While the authors claim that CSG generalizes across tasks, they only test standard supervised image classification. The theoretical link assumes uniform β-stability of SGD and bounded loss, which rarely holds in modern deep nets. It remains unclear whether CSG maintains predictive utility in other regimes such as self-supervised, generative, or multi-label settings.
>
>      **A:** This is a valid criticism, however these assumptions are common for optimizer convergence guarantees in non convex settings [6]. We believe the major hurdle in other regimes is the lack of a strong memorization baseline. For vision models, Feldman and Zhang trained thousands of CIFAR-100 and ImageNet models to establish a baseline. However, no such baseline currently exists for self-supervised, generative, or multi-label settings to rigorously test leave-one-out memorization.

---

> ### Author Response · Authors · 2025-11-21
>
> ### Part (2/2)
>
> **Questions:**
>
> 1. Since CSG is defined as the accumulated input gradient norm, not the derivative of a loss functional over training trajectories, do we have a rigorous reason to treat it as a “gradient of loss with respect to input samples, accumulated over training”? How does this differ from simply tracking cumulative sensitivity?
>
> 	**A:** This is an important question. (a) While one could technically track the full sensitivity vector, the memory overhead is prohibitively large compared to tracking the scalar norm. (b) More importantly, theoretical convergence guarantees for optimizers such as SGD are typically formulated in terms of the gradient norm; thus, tracking the gradient norm aligns more closely with established theory.
>
> 	We have included additional baselines such as TracIn [1], GraNd [2], VoG [3] and SAIMS [4]. VoG is particularly relevant as it tracks the variance of gradients, which serves as a proxy for accumulated sensitivity. TracIn and GraNd also "simply tracking cumulative sensitivity". The results are provided in the table at this **[anonymous ICLR compliant link](https://lively-mushroom-060927110.3.azurestaticapps.net/?tab=mem-sim)**. These experiments demonstrate the consistency of our method across various optimizers and its superiority over these sensitivity-based methods.
>
> 2. The paper asserts that stopping at the peak weight norm matches the minimum validation loss. Can this be proven under general conditions? How robust is this correspondence across architectures, datasets, or optimizers (e.g., Adam, Adagrad, adaptive schedulers)?
>
> 	**A:** We have provided an interactive website with plots for various optimizers here: https://lively-mushroom-060927110.3.azurestaticapps.net/?tab=wd&opt=adagrad&data=ig%2Cvl. Additionally, **[this link](https://lively-mushroom-060927110.3.azurestaticapps.net/?tab=imgnet-csg)** presents results for ViT and ResNet50 on ImageNet using a cosine scheduler (via the `timm` library with augmentation for SoTA training). We found that the input gradient is a more robust indicator than the weight norm for identifying the early stopping point corresponding to the lowest validation loss. As noted in the manuscript (section 4.3 Practical takeaway) , due to the sensitivity of the weight norm, we suggest monitoring both the input gradient and the weight norm.
>
> 3. Do you think the CSG .... please share your intuition on whether and why such generalization might hold.
>
> 	**A:** Theoretically, the results should hold for LLMs or other statistical models that use classification-based pretraining. As discussed prior, the major issue is establishing a valid baseline for these models, since establishing a "leave-one-out" baseline requires training a vast number of models.
>
> 	To build intuition on why this holds, we must consider real-world datasets, which are typically long-tailed [5]. Real datasets can be thought of as a collection of subpopulations where the number of samples in each subpopulation is distributed according to a power law (i.e. long tailed datasets). The reason we expect this to hold for other models stems from how optimization works: early in training, the gradient direction is dominated by the directions that learn the "modes" i.e., subpopulations with the largest number of samples or well represented subpopulations. Once these are learned, the model progressively learns the next largest subpopulations until it is forced to memorize the tail or is unable to learn the tail based on the capacity of the model. As Feldman and Zhang showed, for subpopulations without sufficient samples, memorization is needed for generalization.  This sequential learning is also theoretically shown via the learning time result. Therefore, since this sequential learning dynamic is intrinsic to gradient-based optimization, we expect CSG to remain a valid proxy for memorization across different loss landscapes.
>
> **References:**
>
> [1] Pruthi, G., et al. "Estimating Training Data Influence by Tracing Gradient Descent."
>
> [2] Paul, M., et al. "Deep Learning on a Data Diet: Finding Important Examples Early." (GraNd)
>
> [3] Agarwal, C., et al. "Estimating Example Difficulty using Variance of Gradients." (VoG)
>
> [4] Agiollo, Andrea, Young In Kim, and Rajiv Khanna. "Approximating Memorization Using Loss Surface Geometry for Dataset Pruning and Summarization."
>
> [5] Feldman, V., & Zhang, C. (2020). What neural networks memorize and why: Discovering the long tail via influence estimation.
>
> [6] Ghadimi, Saeed, and Guanghui Lan. "Stochastic first-and zeroth-order methods for nonconvex stochastic programming." SIAM journal on optimization 23, no. 4 (2013): 2341-2368.
>
> [7] Ravikumar, Deepak, Efstathia Soufleri, Abolfazl Hashemi, and Kaushik Roy. "Towards memorization estimation: Fast, formal and free." ICML 2025.

---

### Meta-Review · Area_Chair_W1hj · 2025-12-28

**Summary:**

The paper proposes a state-of-the-art memorization score that matches performance of prior methods but is faster to compute.

Reviewers have appreciated useful performance and simplicity of the metric.

Reviewers maBN and BKPC remarked that the paper's novelty is limited. The contribution is incremental in the light of preexisting works that connect early phase to memorization (https://arxiv.org/abs/2007.00151 ) and even derive memorization metrics (https://arxiv.org/abs/2003.10647). Reviewer BKPC further referenced works such as Variance of Gradients.

Another weakness is that the theory is presented as stronger than it is. The loose bounds do not provide a casual picture that explains the empirical performance.

All in all, it was a borderline paper that has merits but is incremental. I believe it clears the bar for acceptance and I am happy to recommend it for acceptance.

**Reviewer Concerns:**

Reviewers maBN and BKPC raised valid concern that the paper is incremental, which while Authors have commented on, hasn’t been addressed by the rebuttal fully.

Reviewers asked for further experiments, e.g. expanding to more optimizers. This has been addressed sufficiently by the rebuttal.

Reviewers also noted that theory is not that useful in explaining the success of the metric, given that bounds are loose. I agree with Reviewers that the theoretical link is rather loose and the rebuttal hasn’t addressed this sufficiently.

**Reviewer Scores:**

I believe Authors have addressed sufficiently all comments apart from the novelty one and the comment about theory. It is plausible that some reviewers would have raised their scores.

---

### Decision · Program_Chairs · 2026-01-26

Accept (Poster)